# Investigating the effects of artificial baroreflex stimulation on pain perception: A comparative study in no-pain and chronic low back pain individuals

Alessandra Venezia[1] , Harriet-Fawsitt Jones[1], David Hohenschurz-Schmidt[2], Matteo Mancini[3], Matthew Howard[1] and Elena Makovac[1,4]

[1]Department of Neuroimaging, Institute of Psychology, Psychiatry & Neuroscience, King's College London, London, UK
[2]Department of Surgery & Cancer, Imperial College London, London, UK
[3]Department of Cardiovascular, Endocrine-Metabolic Diseases and Aging, Italian National Institute of Health, Rome, Italy
[4]Department of Life Sciences, Division of Psychology, Brunel University London, London, UK

Handling Editors: David Wyllie & Vaughan Macefield

The peer review history is available in the Supporting Information section of this article (https://doi.org/10.1113/JP286375#support-information-section).

*The Journal of Physiology* (side text)

**Abstract figure legend** There is a strict interaction between the autonomic nervous system (ANS), as expression of the cardiovascular state of the body, and pain. Nociception is modulated by 'top-down' descending pain control systems from the brain to the brainstem and spinal cord. Here, we perturbed the ANS by applying a non-painful negative pressure at the level of the carotid bifurcation to stimulate baroreceptors and we evaluated the integrity of a pain modulation mechanism by a conditioned pain modulation (CPM) assessment, in which participants received noxious pressure (used as test stimulus) delivered simultaneously with a continuous pressure stimulation (used as conditioning stimulus) on the thumbnails. We aimed to examine the relationship between ANS reactivity and CPM efficiency in two groups of 'no-pain' and 'chronic low back pain' participants, with a focus on the specific involvement of baroreceptors in this interaction. In our sample, baroreflex activation decreased pain in pain-free participants but increased pain perception in those with chronic pain, indicating that this activation may play a role in the ANS–pain interaction and that is disrupted in chronic pain states. We suggest the potential importance of the baroreflex in pain perception and indicate potential avenues of exploiting these mechanisms to improve treatment of individuals with chronic pain.

This article was first published as a preprint. Venezia A, Jones H-F, Hohenschurz-Schmidt D, Mancini M, Howard M, Makovac E. 2023. Investigating the Effects of Artificial Baroreflex Stimulation on Pain Perception: A Comparative Study in Healthy Participants and Individuals with Chronic Low Back Pain. medRxiv. https://doi.org/10.1101/2023.12.18.23299896

**Abstract**  The autonomic nervous system (ANS) and pain exhibit a reciprocal relationship, where acute pain triggers ANS responses, whereas resting ANS activity can influence pain perception. Nociceptive signalling can also be altered by 'top-down' processes occurring in the brain, brainstem and spinal cord, known as 'descending modulation'. By employing the conditioned pain modulation (CPM) paradigm, we previously revealed a connection between reduced low-frequency heart rate variability and CPM. Individuals with chronic pain often experience both ANS dysregulation and impaired CPM. Baroreceptors, which contribute to blood pressure and heart rate variability regulation, may play a significant role in this relationship, although their involvement in pain perception and their functioning in chronic pain have not been sufficiently explored. In the present study, we combined artificial 'baroreceptor stimulation' in both pressure pain and CPM paradigms, seeking to explore the role of baroreceptors in pain perception and descending modulation. In total, 22 individuals with chronic low back pain (CLBP) and 29 individuals with no-pain (NP) took part in the present study. We identified a differential modulation of baroreceptor stimulation on pressure pain between the groups of NP and CLBP participants. Specifically, NP participants perceived less pain in response to baroreflex activation, whereas CLBP participants exhibited increased pain sensitivity. CPM scores were associated with baseline measures of baroreflex sensitivity in both CLBP and NP participants. Our data support the importance of the baroreflex in chronic pain and a possible mechanism of dysregulation involving the interaction between the ANS and descending pain modulation.

(Received 13 February 2024; accepted after revision 2 September 2024; first published online 9 October 2024)

**Corresponding author** E. Makovac: Brunel University London, Division of Psychology, Department of Life Sciences, Centre for Cognitive and Clinical Neuroscience, Uxbridge, UB8 3PH, UK. Email: elena.makovac@brunel.ac.uk

**Key points**

- Baroreflex stimulation has different effects on pressure pain in participants with chronic pain compared to matched individuals with no-pain.
- Baroreceptor activation decreases pain in participants with no-pain but increases pain perception in participants with chronic pain.
- Baroreflex sensitivity is associated with conditioned pain modulation in both groups of chronic pain and no-pain participants.
- The reactivity of the baroreflex during autonomic stress demonstrated a positive correlation with Pain Trait scores in participants with chronic back pain.

## Introduction

The interplay between the cardiovascular system and pain plays a crucial role in pain regulation. In pain-free individuals, short acute pain increases sympathetic arousal and blood pressure (Kyle & McNeil, 2014), whereas there is variability in the individual response to tonic pain (e.g. muscular pain), with some individuals reporting an increase and others a decrease in sympathetic activation (Fazalbhoy et al., 2012). Conversely, a reduction in pain perception is reported in pain-free participants during spontaneous (Olsen et al., 2013) or induced

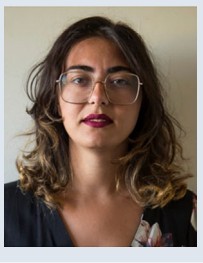

**Alessandra Venezia** is a Research Assistant at the Centre for Neuroimaging Sciences – Institute of Psychiatric, Psychology and Neuroscience at King's College London. She earned her undergraduate degree from University of Trento, Italy, in Cognitive Sciences before moving to the University of Rome 'La Sapienza', Italy, to study for a Master's degree in Neuroscience. Her current research focuses on underpinning brain and spinal cord pain mechanisms using various research methods, including physiology, psychophysics and MR imaging.

(Duschek et al., 2009) high blood pressure (BP), as well as in unmedicated patients with hypertension (Makovac et al., 2020; Saccò et al., 2013). The relationship between hypertension and pain is complex and not yet fully understood. Although hypertension appears to be associated with heightened pain thresholds, studies also indicate a higher prevalence of hypertension in individuals with chronic pain (Parsons et al., 2015). This relationship might be mediated by the baroreceptors, specialised sensory receptors located in the walls of the carotid sinus, which regulate cardiovascular functions by adjusting heart rate and BP to maintain homeostasis (Makovac et al., 2020; Swenne, 2013). Indeed, higher baroreflex sensitivity, assessed by the rate of heart rate response to fluctuations in BP (Swenne, 2013), has been found to correlate inversely with the severity of pain experienced by healthy individuals (Duschek et al., 2007) and with post-surgical pain (Suarez-Roca et al., 2024).

Nociception is also modulated by 'top-down' processes in the brain, brainstem and spinal cord. Descending pain modulation (Youssef et al., 2016a, b) can be triggered by a painful conditioning stimulus, which can suppress incoming nociceptive signals arising from a second stimulus at a different body site. Conditioned pain modulation (CPM) is a widely used paradigm for assessing descending pain modulation in humans, typically performed by applying a painful conditioning stimulus to one part of the body to reduce the pain perception from a test stimulus applied to another part of the body (Nir & Yarnitsky, 2015). A deficient descending pain modulation, as measured by CPM, has been suggested to be one of the mechanisms involved in the persistence of pain. Reduced CPM efficiency has been demonstrated in various chronic pain states, including fibromyalgia (for a meta-analysis on the topic, see O'Brien et al., 2018), chronic migraine (Williams et al., 2019) and chronic temporomandibular pain disorder (Oono et al., 2014). A recent meta-analysis in chronic low back pain (CLBP) patients, however, showed that CPM efficiency in this cohort is controversial, with three published studies reporting significant differences in CPM between CLBP and pain-free controls, whereas four studies did not find a difference (Neelapala et al., 2020). These discrepancies might be the consequence of methodological choices (Nir et al., 2011) or biopsychosocial factors, including pain catastrophising, gender, age and beliefs (Bjørkedal & Flaten, 2012; Popescu et al., 2010). Interestingly, heterogeneity in pathophysiological mechanisms related to CPM has been described, indicating that chronic pain patients can be clustered into different groups based on their CPM profile (Ocay et al., 2022).

An association has been also described between BP and CPM response (Chalaye et al., 2013). The stimulus used in the CPM paradigm elicits not only pain, but also an accompanying cardiovascular response.

Artificial baroreceptor stimulation results in reduced pain sensitivity in both hypertensive and normotensive individuals (Edwards et al., 2003). Baroreceptor stimulation has been also shown to attenuate attentional effects to pain stimuli (Grey et al., 2010). Some pioneering studies have suggested that the pain-attenuating effect of baroreflex stimulation is disrupted in chronic pain (Kennedy et al., 2016), arguing that efficient descending pain modulation depends on effective autonomic regulation and that deficiencies in these mechanisms relate to pain persistence. These data suggest that the autonomic dysregulation described in chronic pain patients (Tracy et al., 2016) alongside the high inter-individual variability of the cardiovascular response to pain (Fazalbhoy et al., 2012) might be related to the CMP response and explain some of the CPM variability described in chronic pain conditions (Neelapala et al., 2020; Ocay et al., 2022).

In the present study, we aimed to examine the relationship between ANS reactivity and CPM efficiency in two groups of NP and CLBP participants, with a focus on the specific involvement of baroreceptors in this interaction. We examined the impact of autonomic perturbation on pain perception using artificial baroreceptor stimulation during a pressure pain paradigm. We also assessed the efficiency of descending pain modulation (i.e. CPM) in changing pain sensations. We hypothesised that the efficiency of the baroreflex would be linked with pain perception and modulation in NP participants, but that this association would be disrupted in CLBP participants.

## Methods

### Ethical approval

The study was approved by King's College London Research Ethics Committee (HR-19/20-14 149) and was conducted in accordance with the *Declaration of Helsinki*, except for registration in a database. All participants were fully informed regarding the experimental procedures and provided their written informed consent prior to participating.

### Participants

In total, 22 CLBP and 29 age- and gender-matched NP participants were recruited via advertisements appearing via social media and the Brixton Therapy Centre in London, UK, to take part in this study. Chronic pain participants were recruited at the Brixton Therapy Centre All participants were right-handed. CLBP was defined as continuous or recurrent episodes of pain in the lower back (with or without pain in a lower extremity) that

persisted or recurred over the past 3 months (Burton et al., 2006; Treede et al., 2019). Other inclusion criteria for CLBP participants were: (1) age between 18 and 65 years; (2) no structural pathology; and (3) under stable or no pharmacological management.

Exclusion criteria included a history of brain injuries, hypertension, neurological or psychiatric disease, and alcohol or drug abuse. Additional cardiovascular exclusion criteria included: personal or family history of hypertension; smoking more than five cigarettes a day; body mass index $>30$ kg m$^{-2}$; a history of orthostatic hypotension or carotid hypersensitivity. At the beginning of each visit, participants were tested for drug use and alcohol consumption using a urine drug screen and alcohol breathalyser test, respectively. To further exclude possible alterations of the ANS, a Valsalva manoeuvre was performed at the beginning of each session, whereby a decrease in systolic pressure of $>20$ mmHg was defined as abnormal ANS functioning and specified as an exclusion criterion. Prior to each session, participants were required to abstain from alcohol for 24 h, non-steroidal anti-inflammatory drugs and paracetamol for 12 h, tobacco and nicotine-containing products for 4 h, and to limit caffeine intake to a maximum of one caffeinated drink.

### Experimental setup for baroreceptor stimulation

A bespoke baroreceptor stimulating device was developed by engineers at the Department of Neuroimaging at King's College Londo. The device used two individual cuffs to apply non-painful negative pressure ($-60$ mmHg) for efficacious baroreceptor stimulation (ACTIVE condition) and $-20$ mmHg stimulation (SHAM condition) applied to the neck at the location of the carotid bifurcation. The device has been used safely in previous studies with healthy individuals (Basile et al., 2013; Makovac et al., 2018; Makovac et al., 2015) to deliver reliable modulation of peripheral (cardiovascular) and central (neuronal) mechanisms (Makovac et al., 2015, 2018).

### General overview of the experimental procedures

Participants took part in one experimental session which lasted ∼2 h (Fig. 1). At the beginning of each experimental session, participants were asked to rate the intensity level of their current pain (pain state) and their average pain levels during the past year (pain trait) (Davis & Cheng, 2019), using a numerical rating scale from 0 to 100 (0 indicating no pain, 100 indicating worse imaginable pain).

Next, a pain thresholding session was conducted to identify participants' individual pressure pain thresholds. For a thorough characterisation of the baroreflex activity, participants' ANS activity was tested at rest (REST condition), during autonomic stress induced by a tonic cold pain stimulus (STRESS condition), or in response to baroreceptor stimulation (ACTIVE condition). Across all the conditions, participants were blind to the ACTIVE and SHAM stimulation. Finally, participants completed two experimental tasks, both of which involved pressure stimulation. First, a pain task, during which noxious pressure was delivered coincident with simultaneous ACTIVE or SHAM baroreceptor stimulation. The second was a CPM experiment in which pressure stimulation was delivered to participants' right thumbnails, under two conditions; a test stimulus ('pressure only' baseline condition), or with simultaneous bilateral pressure stimulation ('CPM' condition), During

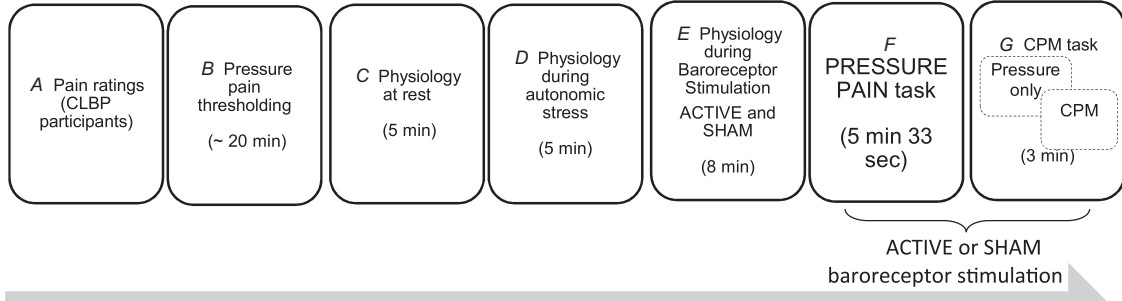

**Figure 1. Graphical representation of the experimental session**
*A*, participants rated their current pain and their average pain during the past year. *B*, pain thresholding: individual pressure pain thresholds and a moderate pain level corresponding to a 70/100 VAS were determined. *C* and *D*, baseline (resting) physiology and physiology in response to autonomic stress (cold pain), each measured for 5 min. *E*, physiology measurement during ACTIVE and SHAM baroreceptor stimulation. *F* and *G*, PRESSURE PAIN and the CPM tasks. Participants rated the intensity of pressure pain presented on the right thumbnail (PRESSURE PAIN task) or simultaneously on the left and right thumbnail (CPM task). In both (*F*) and (*G*) pressure pain stimuli were presented coincident with either ACTIVE or SHAM baroreceptor stimulation.

the CPM condition, pressure stimuli were delivered to the right thumbnail (test stimulus) and left thumbnail (conditioning stimulus). These stimuli were both presented with either ACTIVE or SHAM baroreceptor stimulation.

**Pain ratings (Fig. 1A).** Following a recent study highlighting the importance of measuring stable and transitory pain levels in chronic pain patients (Davis & Cheng, 2019), CLBP participants were asked to rate their usual levels of low back pain (pain trait) and their current levels of pain (pain state). Pain trait was defined as the average intensity of low back pain experienced over the past year, rated on a scale ranging from 0 (no pain) to 100 (the most severe imaginable pain). Pain state, on the other hand, referred to the specific level of lower back pain experienced on the day of the experimental testing, also measured on a scale from 0 to 100 .

**Pressure pain thresholding (Fig. 1B).** Sensory thresholding for pressure pain was performed using an established ascending sequential staircase paradigm (Jackson et al., 2020). We used a custom-designed pain pressure probe, which provided perpendicular force to the thumbnail. The ascending staircase paradigm started with an initial force of 131.29 kPa applied to the nail surface for 2 s. Incremental increases of 18.06 kPa were subsequently applied. Participants indicated when they first felt pain (their pain detection threshold), as well as when they reached a moderate to severe level of pain, corresponding to 70/100. The maximum force that could be applied was 726.82 kPa. Next, participants received 15 pressure stimulations, delivered in a pseudo-random order. The minimum pressure delivered corresponded to each participant's pain threshold and the maximum pressure to their 70/100 rating. Three intermediate pressure pain levels were also delivered, each equally spaced from one another. Participants rated each stimulation immediately afterwards. Participants' final 70/100 threshold was then calculated based on a regression analysis derived from the 15 stimulations (Jackson et al., 2020).

**Physiological measurements at baseline and in response to autonomic stress (Fig. 1C and D).** Continuous heart rate (HR) and BP at rest and in response to autonomic stress were measured over 5 min using a CareTaker device (https://caretakermedical.net/en-gb/home-2/), placed on the ring finger of the right hand. Cold pain stimulation, used to induce autonomic stress, was delivered via a locally-developed aluminium probe (dimensions: width 4 cm, length 20 cm), attached to the volar surface of the left forearm, through which cold water at 4°C was constantly circulated by means of two chillers (Fig. 2); for a similar procedure, see also Makovac et al. (2019). During these

recordings, participants were placed in a relaxed, supine position and instructed not to move.

**Experimental paradigm assessing the effect of baroreceptor stimulation on physiology (Fig. 1E).** Participant's HR was measured during both the ACTIVE and SHAM conditions to assess the impact of baroreceptor stimulation without any painful stimulation. To examine the physiological response to baroreceptor stimulation, a pulse oximeter was used. The baroreceptor stimulation lasted for 8 s, followed by a 12 s inter-trial interval, as per the durations used in previous studies (Makovac et al., 2015, 2018), which are known to cause activations in the ANS and subsequent baroreceptor recovery. In total, eight ACTIVE and eight SHAM stimulations were delivered, for a total duration of the experimental task of 8 min.

**Experimental paradigm assessing the effect of baroreceptor stimulation on pressure pain (Fig. 1F).** In the PRESSURE PAIN task, we examined the effect of the activation of baroreceptors on the perception of pressure pain, delivered simultaneously with either ACTIVE or SHAM artificial baroreceptor stimulation. Negative neck suction started 6 s before the painful stimuli to facilitate baroreceptor activation (Fig. 3). Pressure pain (2 s in duration) was delivered to the right-hand thumbnail, at the end of the stimulation interval. The total duration of baroreceptor stimulation was 8 s. Each pressure pain

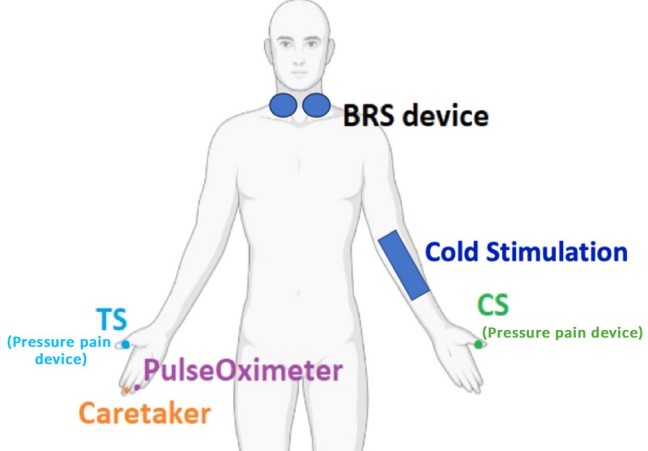

**Figure 2. Graphical representation of the experimental setup**
Participants remained connected to the neck suction device throughout the duration of the experiment. The activation of the neck suction (baroreceptor stimulation) was synchronised with the presentation of pain stimuli in all experimental paradigms. Cold stimulation via an aluminium probe was used during the autonomic stress only and removed from participant's arm after the stimulation, to avoid carry-over effects. Pressure pain (test stimulus; TS) was delivered to the right thumbnail, whereas the conditioning stimulus (CS) was delivered to the left thumbnail. Continuous HR and BP were collected from the middle and ring fingers, respectively. [Colour figure can be viewed at wileyonlinelibrary.com]

stimulus was followed by a 12 s rating interval, which also allowed for the recovery of the baroreceptors. Each trial lasted 20 s. Overall, participants rated eight pressure + ACTIVE and eight pressure + SHAM trials, for a total duration of the experimental run of 5 min and 33 s.

**Experimental paradigm assessing the effect of baroreceptor stimulation on CPM (Fig. 1***G***).** In the CPM task, participants were presented with two types of trials: 'pressure only' and 'CPM'. During 'pressure only' trials, pressure pain was used as a test stimulus, delivered to the right-hand thumbnail. Each trial started with a fixation cross of 2000 ms in duration (Fig. 4). Participants then received one test stimulus (2000 ms in duration), followed by a visual analogue scale (VAS) rating period (11,000 ms in duration) during which participants rated the intensity of their pain on a scale from 0 to 100.

During 'CPM' trials, the same test stimulus was repeated with a simultaneous conditioning stimulus, which consisted in a continuous pressure stimulation (4000 ms in duration) applied to the left thumbnail. Participants were instructed to rate the perceived pain in response to the test stimulus and to ignore the pain elicited by the conditioning stimulus. During both the 'pressure only' and 'CPM' trials, painful stimulations were delivered with a simultaneous ACTIVE or SHAM baroreceptor stimulation, which were randomly delivered. The baroreceptor stimulation reached its maximum negative pressure at the onset of the conditioning stimulus. The test stimulus was delivered 3000 ms following the onset of the baroreceptor stimulation, and 1000 ms following the onset of the conditioning stimulus (Fig. 4).

Each trial lasted 19 s. Participants completed a total of eight trials: two 'pressure only' trials and two 'CPM' trials, each with ACTIVE or SHAM baroreceptor stimulation, randomly delivered. The task lasted 2.53 min.

### Statistical analysis

Normality of the variables was examined using Shapiro–Wilks tests. Non-normally distributed variables were logarithmically transformed before proceeding with further analyses. All data are expressed as the mean ± SD. Parametric tests were used when normality was achieved; otherwise, non-parametric tests were selected. Age and gender were included as covariates in all analyses. $P < 0.05$ was considered statistically significant. Data analyses were conducted using SPSS, version 23.0 (IBM Corp., Armonk, NY, USA).

**Pre-processing of physiological data.** Interbeat intervals were identified for data collected by means of the CareTaker and photoplethysmography (PPG). CareTaker was initially calibrated based on each participant's resting BP. Subsequently, interbeat intervals were automatically detected using CareTaker. PPG interbeat intervals were calculated with an in-house Matlab 9.7 (R2019b) script (https://www.mathworks.com/products/matlab.html) running on a Linux platform. Interbeat intervals values were visually inspected, and potential artifacts removed manually. Interbeat intervals were then further corrected by using a threshold-based artifact correction algorithm, adapted from the software Kubios HRV Standard, version 3.0.2 (Tarvainen et al., 2014). The algorithm compared each interbeat interval to a local median calculated from

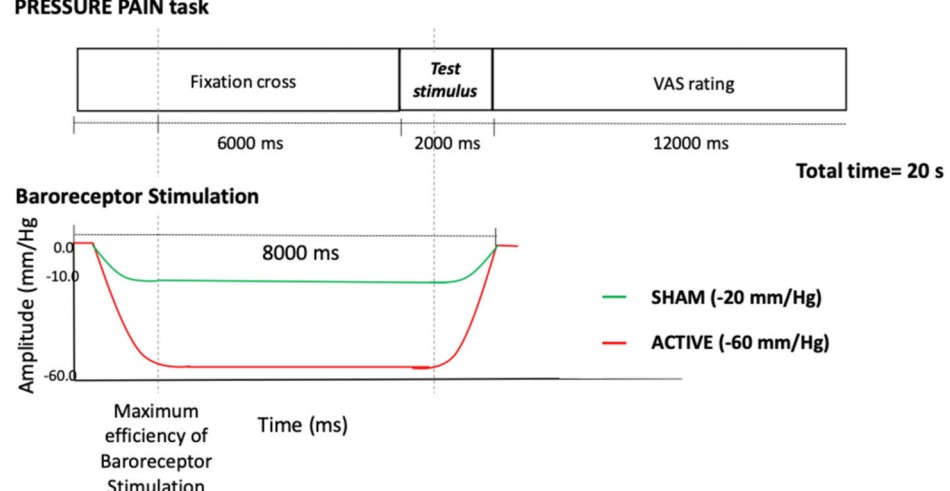

**Figure 3. Graphical representation of one experimental trial during the pressure pain paradigm**
Pressure pain was administered for 2 s on the right-hand thumbnail, either during ACTIVE or SHAM baroreceptor stimulation. Negative neck suction was initiated 6 s prior to the painful stimuli to activate the baroreceptors. Following each pressure pain stimulation, a 12 s rating (VAS) interval allowed for baroreceptor recovery. [Colour figure can be viewed at wileyonlinelibrary.com]

five nearby heartbeats. If the interbeat interval deviated from the local median beyond a specified threshold, it was considered an artifact and replaced with the median value. Different threshold values (0.45, 0.35, 0.25, 0.15 or 0.05 s) were used in a sequential manner, from conservative to liberal. The appropriate correction threshold was selected based on the severity of the individual artifact and followed the procedure described in the Kubios manual (https://www.kubios.com/hrv-preprocessing). First, we identified beat intervals requiring corrections. If present, we selected the minimum correction level, rectifying the abnormal beat without excessively altering the remaining data. Finally, interbeat interval intervals were converted to HR values to facilitate data interpretation.

**Baroreflex sensitivity index.** The baroreflex sensitivity index (BSI) is an index of baroreflex responsiveness, indicating the degree of control of the baroreflex over HR in response to changing BP levels (La Rovere et al., 2008). The BSI was calculated from HR and BP data collected during 5 min rest and cold conditions, using the sequence method, described in Laude et al. (2004). Briefly, the method is based on the identification of three or more consecutive beats in which progressive increases/decreases in systolic BP are followed by progressive lengthening/shortening of RR intervals. The threshold values for including beat-to-beat systolic BP and interbeat interval changes in a sequence were set at 1 mmHg and 6 ms, respectively. The slope of the linear regression line fitted to each sequence was calculated, representing the BSI. The average slope value was used as a measure of BSI.

**Analysis of the autonomic activity during autonomic stress.** The effect of autonomic stress (cold pain) on BSI and interbeat interval values was evaluated by means of a 2 × 2 ANOVA, with cold (ON, OFF) and group (NP, CLBP) as main factors. Age and gender were used as covariates of no interest.

**Analysis of autonomic activity in response to *baroreceptor stimulation*.** To investigate the effect of baroreceptor stimulation on physiology, mean HR values were calculated from 8 s time windows, corresponding to the baroreceptor stimulation window. (As a result of a malfunction in our experimental setup, we were only able to evaluate the effect of baroreceptor stimulation on mean HR values because we could not compute the BSI under these conditions.) The effect of baroreceptor stimulation on HR intervals was investigated by means of a 2 × 2 ANOVA, with condition (ACTIVE, SHAM) as the main within-subject factor and group (CLBP, NP) as a between-subjects factor.

We then derived the HR_active score (Table 1), defined as the difference in HR between the ACTIVE and SHAM condition (HR_active = HR Active – HR Sham). Here, negative numbers indicated a decrease in the HR during ACTIVE compared to SHAM conditions. HR_active scores were used in correlational analyses to explore the association between the physiological reaction to baroreceptor stimulation and behavioural variables (Table 1).

**Analysis of behavioural data from PRESSURE PAIN paradigm.** In the PRESSURE PAIN task, the effect

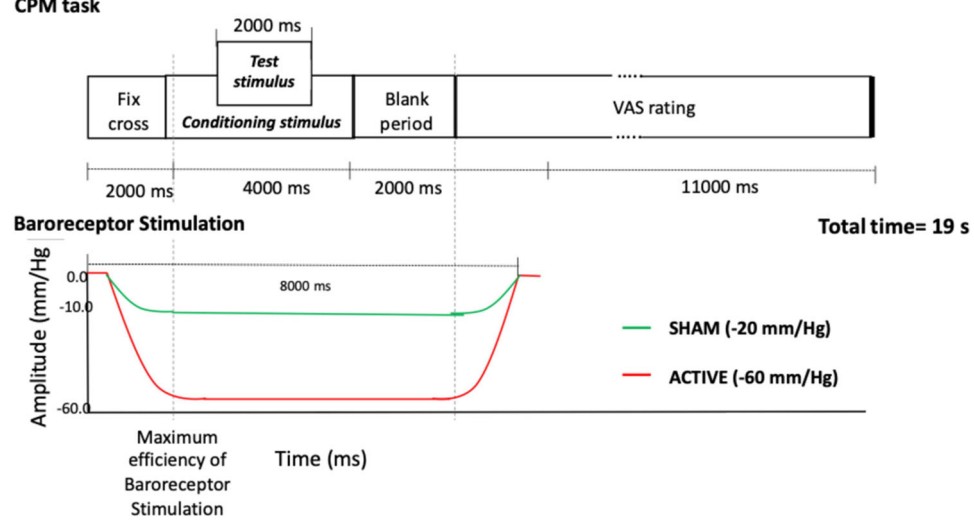

**Figure 4. Graphical representation of one experimental trial during the CPM paradigm**
During 'CPM' trials, participants received a pressure pain test stimulus either on its own ('pressure only' trials), or concurrently with a conditioning stimulus (CPM trials). Both trials involved random delivery of painful stimulations accompanied by either ACTIVE or SHAM baroreceptor stimulation. [Colour figure can be viewed at wileyonlinelibrary.com]

**Table 1. Description of physiological and behavioural measures**

| Name | Description | Unit |
|------|-------------|------|
| *Physiological measures* | | |
| BSI_rest | Baroreceptor sensitivity index and HR at rest, in supine position | Mean slope |
| HR_rest | | Milliseconds |
| BSI_stress | Baroreceptor sensitivity index and HR during autonomic stress (cold pain), in supine position | Mean slope |
| HR_stress | | Milliseconds |
| HR_active | HR during artificial baroreceptor stimulation, in supine position | Milliseconds |
| *Behavioural pain measures* | | |
| PRESSURE_active | The effect of ACTIVE or SHAM baroreceptor stimulation on pressure pain perception | VAS |
| PRESSURE_sham | | VAS |
| Delta_PRESSURE | The difference in pressure pain perception derived by subtracting the rating of pressure pain in SHAM – ACTIVE condition | Delta VAS |
| CPM_active | The activity of the descending pain modulation during ACTIVE or SHAM baroreceptor stimulation, defined as the difference in subjective pain rating between the 'Pressure only' and the 'CPM' condition | Delta VAS |
| CPM_sham | | Delta VAS |

For technical reasons, continuous BP was not measured during baroreceptor stimulation, thus not allowing for calculations of the BSI_active index.

of baroreceptor stimulation on pressure pain was investigated with a 2 × 2 ANOVA, with condition (ACTIVE, SHAM) as the main within-subject factor and group (CLBP, NP) as between-subject factor.

Next, differential scores were calculated for pain intensity ratings to facilitate correlational analyses (Table 1). Delta_PRESSURE was calculated as the difference in VAS pain ratings between the SHAM and ACTIVE conditions. Here, negative values indicated a decrease in perceived pain intensity during the ACTIVE baroreceptor stimulation compared to the SHAM stimulation, whereas positive values indicated an increase in perceived pain intensity during the same condition.

**Analysis of behavioural data from the CPM paradigm.** In the CPM paradigm, we adopted a 2 × 2 × 2 ANOVA to explore the effect of the within-subject factors pain condition ('pressure only', 'CPM') and baroreceptor stimulation (ACTIVE, SHAM) and the between-subject factor group (CLBP, NP).

Next, differential CPM scores were calculated separately for the ACTIVE and SHAM conditions, using the formula: CPM_active/sham = VAS ('CPM') – VAS ('pressure only') (Table 1). Here, negative values indicate a decrease in perceived pressure pain during the CPM condition, whereas positive numbers indicate an increase in perceived pressure pain.

**Correlational analyses.** We investigated the association between physiology measures (systolic and diastolic BP, BSI_rest/HR_rest, BSI_stress/HR_stress, HR_active), clinical scores (pain state and trait) and our behavioural measures indicating the VAS rating during painful stimulation (CPM_active, CPM_sham, PRESSURE_active, PRESSURE_sham, Delta_PRESSURE) (Table 1). We computed correlation analyses, namely, Pearson's r (for normally distributed data) or Spearman's test (when normality of the data was not achieved by means of log transformation). Correlations were computed within both the whole sample of NP and CLBP participants (to identify associations which were present independently of the chronic pain condition), as well as within the two groups separately, to seek different patterns of associations in NP and CLBP participants.

## Results

### Sample characteristics

Participants' demographic and clinical characteristics are detailed in Table 2. CLBP and NP participants were matched for age and gender (Table 3); however, there was a trend toward younger age in the NP group compared to CLBP ($P = 0.080$). CLBP and NP participants did not

**Table 2. Demographical profile of CLBP and age and gender-matched NP participants**

| Demographic measures | NP ($n = 29$) | CLBP ($n = 22$) | P value |
|---|---|---|---|
| Age (years), mean $\pm$ SD | 33.7 $\pm$ 10.8 | 39.2 $\pm$ 11.7 | $P = 0.080$ |
| Gender (females) | 13/29 | 14/22 | $\chi2 = 0.06$, $P = 0.821$ |
| Education (years), mean $\pm$ SD | 18.25 $\pm$ 2.1 | 18.52 $\pm$ 2.5 | $P = 0.696$ |
| Body mass index (kg m$^{-2}$), mean $\pm$ SD | 24.4 $\pm$ 3.5 | 23.6 $\pm$ 3.9 | $P = 0.581$ |
| Alcohol consumption (units week$^{-1}$), mean $\pm$ SD | 4.1 $\pm$ 3.9 | 5.8 $\pm$ 3.7 | $P = 0.169$ |

**Table 3. Summary of pain-related measures in the group of NP and CLBP participants**

| Pain-related measures | NP ($n = 29$) | CLBP ($n = 22$) | P value |
|---|---|---|---|
| Pressure pain threshold left hand (kPa) | 347.7 (46.55) | 317.7 (42.83) | $P = 0.369$ |
| Pressure pain threshold right hand (kPa) | 360.3 (48.65) | 393.9 (53.27) | $P = 0.399$ |
| Disease duration (years) | – | 8.5 (6.2) | – |
| Pain trait (average low back pain levels during the past year, 0–100) | – | 59.3 (16.3) | – |
| Pain state (during the experimental session, 0–100) | – | 38.3 (21.2) | – |
| Interference of pain with mood (VAS 1–10) | – | 5.91 (2.71) | – |
| Interference of pain with enjoyment of life (VAS 1–10) | – | 5.09 (2.60) | – |
| Interference of pain with daily activities (VAS 1–10) | – | 5.45 (3.17) | – |
| Effectiveness of pain relief methods used (VAS 0–10) | – | 5.13 (2.32) | – |

NP, no-pain participants; CLBP, chronic low back pain participants. Data represent the mean ($\pm$ SD).

differ in their pressure pain thresholds in either left or right hand (Table 3). CLBP participants' levels of pain during the past (pain state) year and on the day of testing (pain trait) are reported in Table 3. Most chronic pain participants reported weekly pain episodes, which interfered with their work and sleep (Table 3). We did not observe significant differences in BP values between CLBP and NP participants (systolic BP, NP $= 120.6 \pm 9.7$, CLBP $= 119.4 \pm 7.7$, $P = 0.621$; diastolic BP, NP $= 69.1 \pm 7.9$, CLBP $= 69.3 \pm 9.6$, $P = 0.958$).

### Effect of autonomic stress on ANS activity

Two individuals from the NP group were excluded from the analysis as a result of a high number of artifacts ($>30\%$ of heart beats) in the CareTaker data recorded during cold stimulation. Accordingly, BSI analyses during cold stimulation were performed in 22 CLBP and 24 NP participants. BSI values at rest and during cold were non-normally distributed ($W = 0.91$, $P < 0.002$, and $W = 0.85$, $P < 0.001$, respectively). Logarithmic transformation was applied to achieve normality. We did not observe a main effect of the factor Cold ($F_{1,42} = 1.0$, $P = 0.323$) of the factor group ($F_{1,42} < 1$), nor of the cold $\times$ group interaction ($F_{1,42} < 1$) on BSI, indicating that the autonomic stress of cold stimulation itself did not have a significant effect on BSI, regardless the participant group.

### Effect of baroreceptor stimulation on ANS activity

HR values at rest and during active baroreceptor stimulation were non-normally distributed ($W = 0.936$, $P = 0.016$, and $W = 0.937$, $P = 0.016$, respectively). Logarithmic transformation was applied to achieve normality. Overall, when examining the effect of baroreceptor stimulation on HR, we did not observe a main effect of baroreceptor stimulation ($F < 1$), nor a baroreceptor stimulation x group interaction ($F < 1$). We observed a main effect of the group [$F_{1,41} = 2.24$, $P = 0.049$), driven by lower HR in CLBP compared to NP (mean $\pm$ SD: CLBP $= 57 \pm 5$ bpm, NP $= 61 \pm 8$ bpm).

### Effect of baroreceptor stimulation on pressure pain ratings

The distributions were significantly non-normal for the ratings of Pressure pain during ACTIVE ($W = 0.94$, $P = 0.009$) and SHAM ($W = 0.95$, $P = 0.039$) baroreceptor stimulation, according to Shapiro–Wilks tests. Logarithmic transformation was applied to achieve normality. A 2 $\times$ 2 ANOVA revealed a significant baroreceptor stimulation $\times$ group interaction ($F_{1,49} = 6.50$, $P = 0.014$) (Fig. 5A and B). This interaction was driven by a difference in the effect of baroreceptor stimulation on pain perception amongst the two groups of no-pain and chronic pain, where NP participants reported a reduction in perceived pressure pain, whereas CLBP

participants reported an increase in pressure pain (*t* test on Delta_PRESSURE, $t_{49} = 2.6$, $P = 0.013$). We did not observe a main effect of baroreceptor stimulation ($F < 1$), nor for the group belonging ($F_{1,49} = 1.45$, $P = 0.233$).

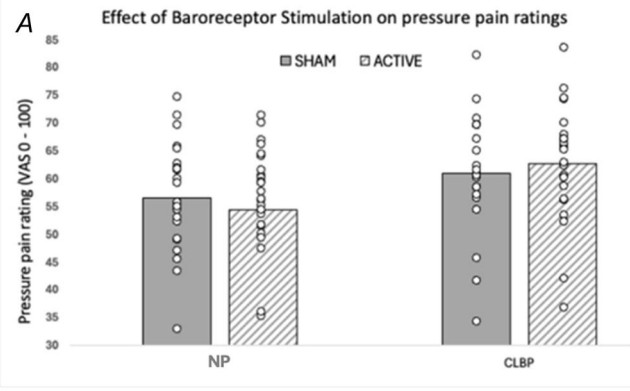

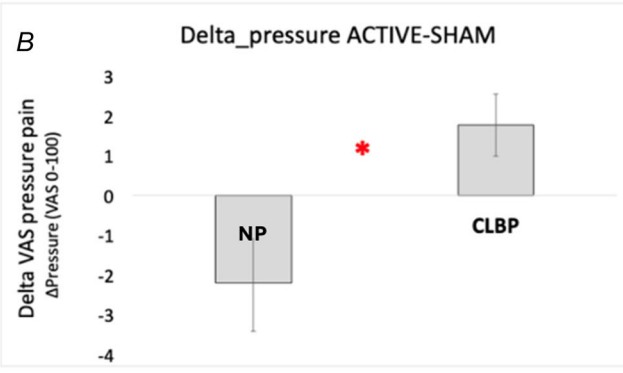

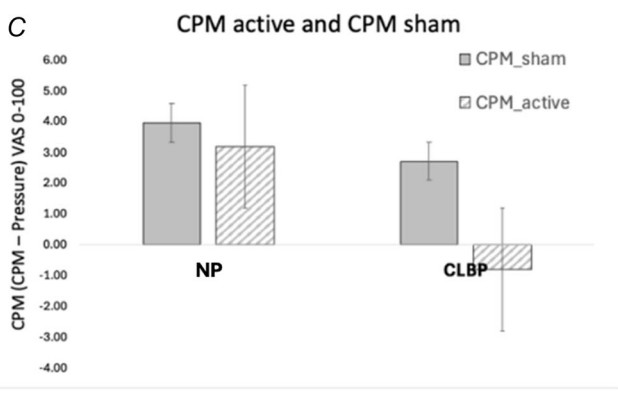

**Figure 5. Effect of ACTIVE and SHAM baroreceptor stimulation on pressure pain and CPM**

*A* and *B*, effect of ACTIVE and SHAM baroreceptor stimulation on perception of pressure pain in NP (*n* = 22) and CLBP participants (*n* = 29) (mean ± SEM). Circles represent individual data points. ACTIVE baroreceptor stimulation reduced the perceived pressure pain in NP and increased pain perception in CLBP participants. *B*, delta pressure score in NP and CLPB groups. *C*, CPM during ACTIVE and SHAM conditions in NP and CLBP groups. [Colour figure can be viewed at wileyonlinelibrary.com]

### Effect of baroreceptor stimulation on CPM

We did not observe an effect of baroreceptor stimulation ($F < 1$), nor of the condition ('pressure only', 'CPM') ($F < 1$) or group ($F < 1$). Similarly, no effect of the baroreceptor stimulation × group interaction ($F_{1,47} = 1.9$, $P = 0.175$) of the baroreceptor stimulation × condition ($F_{1,47} = 1.4$, $P = 0.244$) or of the condition × group interaction ($F < 1$) was observed, indicating no effect of baroreceptor stimulation on the CPM responses in both groups of NP and CLBP (Fig. 5*C*, represented as Delta scores for illustrative purposes).

### Association between BSI_rest and BSI_stress and pressure pain and CPM ratings

In the CPM task, an association was observed across the whole group of NP and CLBP participants between the CPM_active and BSI_rest, indicating that individuals with the highest baroreceptor sensitivity at rest showed the strongest reduction in pain during CPM_active ($r = -0.43$, $P = 0.002$). The same direction of effect was evident during the CPM_sham condition, without reaching statistical significance (Fig. 6).

### Association between HR_active and pressure pain and CPM ratings

Five NP and two CLBP participants were excluded from the analysis due to a high number of artifacts (>30% of heart beats) in the pulse oximeter data recorded during baroreflex stimulation. Accordingly, BSI analyses during cold stimulation were performed in 20 CLBP and 24 NP participants.

We observed a negative association between HR_active changes and the amount of pressure pain modulation induced by ACTIVE baroreceptor stimulation during the pressure pain task (Fig. 7). Here, a decrease in HR was associated with lower ratings of pressure pain during ACTIVE stimulation ($r = -0.46$, $P = 0.043$) in CLBP participants. The same effect was observed in NP participants; however, it did not reach statistical significance ($r = -0.29$, $P = 0.167$). In the CPM task, no association was observed between the HR_active and CPM scores (all $P > 0.1$).

### Association between pain state and trait in CLBP participants and ANS activity

We observed an association between the clinical levels of pain trait and pain state and BP. Specifically, the pain state was negatively associated with diastolic BP, where CLBP participants with higher BP reported lower pain on the day of the testing. Similarly, pain trait was negatively

associated with systolic BP, where individuals with higher BP reported lower levels of pain during the past year (Fig. 8).

A positive association was observed between the pain trait and BSI_stress, where individuals with higher levels of BSI during cold reported higher levels of low back pain during the past year (Fig. 9).

## Discussion

ANS activity and BP can influence pain perception (Fillingim et al., 1998; Pfleeger et al., 1997; Saccò et al., 2013). Both animal studies (Maixner et al., 1981) and experimental studies with manipulation of

baroreceptors in humans (Rau et al., 1994) suggest that the baroreflex accounts at least in part for cardiovascular–pain association. Although the nociceptive and the ANS inter-act at peripheral, spinal cord, brainstem and forebrain levels, the mechanisms underlying this relationship are not fully described. In the present study, we investigated the association between the baroreflex, pain from a noxious pressure stimulus and descending pain modulation in a group of no-pain and chronic pain participants. We used artificial baroreceptor stimulation to explore whether the baroreflex differentially modulates pain perception in NP and CLBP participants. In our CLBP group, autonomic activity at rest, as indicated by resting BP, showed a consistent association between BP and pain perception. However, when the baroreflex was stimulated, via baroreceptor stimulation or autonomic stress, the association between the baroreflex and pain involved distinct mechanisms in participants with chronic pain and with no pain, indicating that the reactivity of

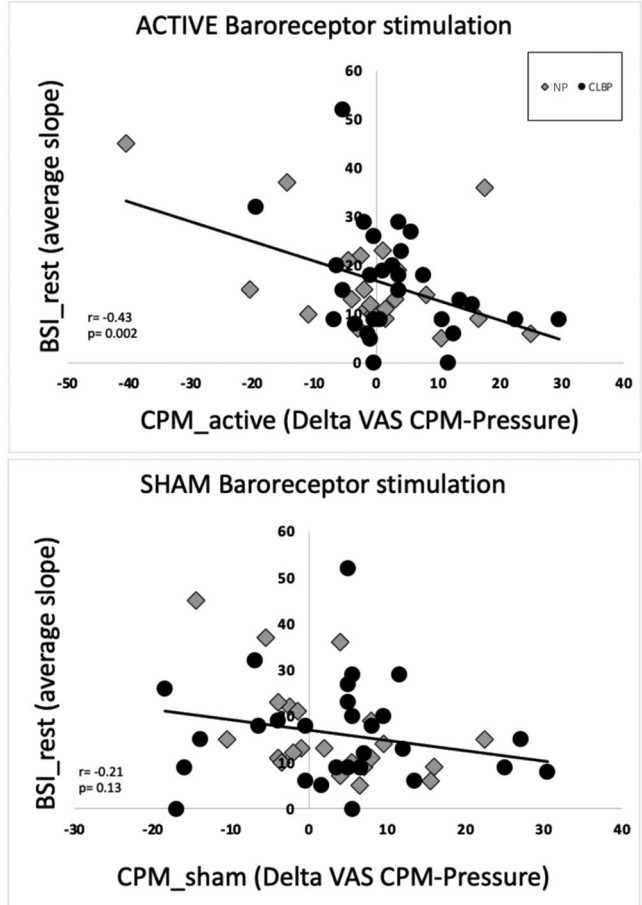

**Figure 6. Association between BSI_rest and CPM ratings**
Negative association between BSI_rest and (upper) CPM_active and (lower) CPM_sham across both NP (*n* = 29) and CLBP (*n* = 22) participants. Only the association between baroreceptor sensitivity and CPM during ACTIVE baroreceptor stimulation reached statistical significance. BSI_rest = Baroreceptor Sensitivity Index at rest; CPM_active/sham = activity of the descending pain modulation during ACTIVE or SHAM baroreceptor stimulation. Note that negative CPM (active/sham) values indicate a decrease in perceived pressure pain during the CPM condition, whereas positive numbers indicate an increase in perceived pressure pain.

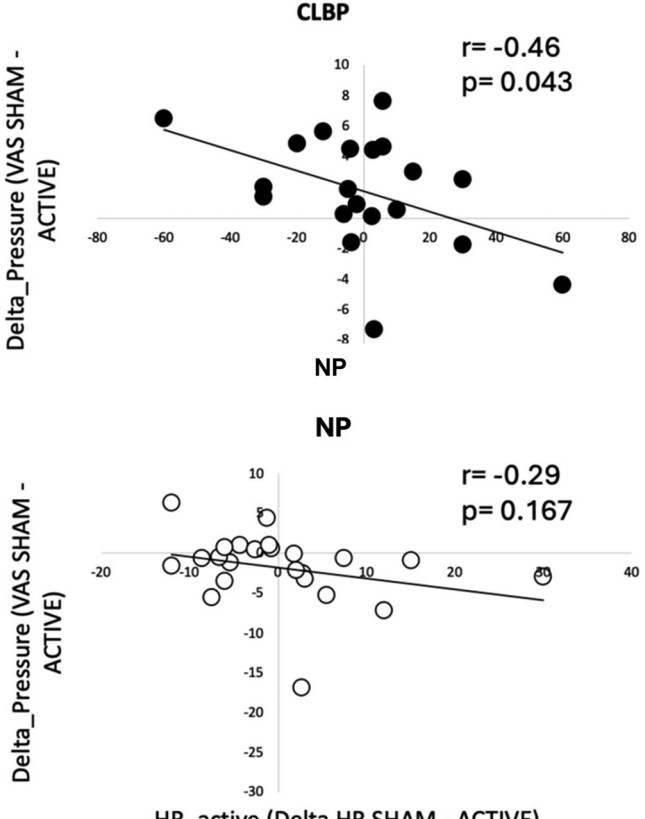

**Figure 7. Association between HR_active and pressure pain ratings in NP and CLBP**
Association between HR_active and pressure pain ratings in NP (*n* = 24) and CLBP (*n* = 20). A negative association was observed between an increase in HR during ACTIVE baroreflex stimulation and a decrease in pressure pain ratings in CLBP participants (upper). The same effect was evident in NP participants; however, it did not reach statistical significance (lower).

the baroreflex might be compromised in chronic pain conditions.

We identified a negative association between BP and measures of low back pain (both state and trait pain) in our chronic pain participants, supporting theories indicating that higher BP levels may be associated with higher pain threshold and tolerance (Makovac et al., 2020). Higher diastolic BP was associated with reduced back pain during the testing session (pain state). Similarly, higher systolic BP was associated with lower levels of pain during the past year (pain trait). The discovery of this relationship intact in CLBP participants partially disagrees with previous studies that suggested the relationship between BP, thermal pain thresholds and pain tolerance is disrupted in chronic pain (Chung et al., 2008). However, pain threshold and tolerance in response to acute experimental pain might not adequately capture the mechanisms of pain-ANS association in individuals who are in ongoing pain. Alternatively, a preserved association between BP and clinical pain may be a result of the otherwise relatively healthy cardiovascular system of the participants in our cohort because only CLBP participants without ANS dysregulation (as

assessed, for example, by the Valsalva manoeuvre) were selected. Close control of cardiovascular risk may have precluded the inclusion of chronic pain participants in whom an association between BP, specifically, baroreflex sensitivity and clinical pain may be compromised. Future studies should aim to enhance our understanding of these mechanisms by incorporating chronic pain participants with elevated cardiovascular risk. Early studies by Bruehl et al. (1998) demonstrated that, although patients with chronic pain conditions lasting 6–14 months preserve the inverse relationship between pain and BP, this relationship turns positive in patients with pain durations exceeding 28 months, indicating a progressive disruption of this mechanism as pain chronicity advances. A critical unresolved question, which might be crucial for prevention of pain chronification, is the specific point at which the association between pain and BP begins to deviate or become disrupted, and the factors that contribute to this disruption.

Further analyses revealed an alteration in the association between the baroreflex and pain in CLBP participants, in conditions where the baroreflex was activated either by autonomic stress or by artificial baroreceptor stimulation. The reactivity of the baroreflex during autonomic stress demonstrated a positive correlation with pain trait scores, indicating that individuals with the highest increase in BSI during autonomic stress reported the most severe pain experienced over the past year. In pain-free individuals, physiological (i.e. cold stimulation) or psychological stress is known to reduce BSI (Gianaros et al., 2012), possibly as a result of inhibitory control originating from the hypothalamus and targeting the nucleus tractus solitarius (Daubert et al., 2012). Similarly, another study by Sabharwal et al. (2004) found that cold exposure reduced baroreflex

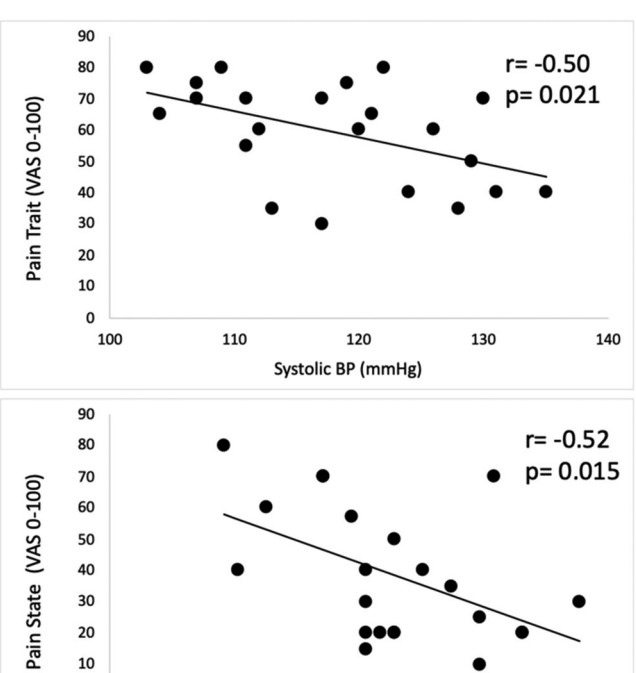

**Figure 8. Association between baseline BP and pain trait and state measures in CLBP participants**
Association between baseline BP and pain trait and state measures in CLBP participants (*n* = 22). A negative association was observed between pain state and diastolic BP, as well as pain trait and systolic BP.

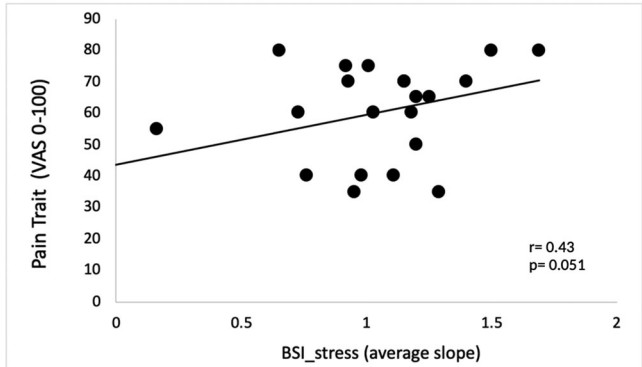

**Figure 9. Positive association between pain trait and BSI_stress in CLBP participants**
Positive association between pain trait and BSI_stress in CLBP participants (*n* = 22). Individuals with the highest increase in BSI during autonomic stress reported the highest pain during the past year.

sensitivity in rats (Sabharwal et al., 2004). This reduction is probably a result of activation of the sympathetic nervous system, which can cause vasoconstriction and an increase in BP. Therefore, transient reductions in BSI during stress are a normal reaction and probably part of a well-functioning homeostatic mechanism. Previous studies have shown that individuals with chronic pain exhibit blunted cardiovascular reactivity to stress (Reyes del Paso & de la Coba, 2020) and orthostatic challenges (Contreras-Merino et al., 2022), indicating that the inhibitory effect of stress on the baroreflex might be dysregulated in these individuals. Consequently, the influence of the baroreflex on descending pain modulation pathways may shift towards pain facilitation rather than pain inhibition during stress. Our results are in line with this hypothesis. Indeed, in the present study, baroreflex stimulation reduced perceived pressure pain in NP participants, whereas the opposite pattern was observed in CLBP participants. Furthermore, a positive correlation was observed in the NP group between the impact of baroreflex stimulation on pain and its effect on physiological responses. On the other hand, this relationship was disrupted in CLBP participants, aligning with previous studies involving chronic pain individuals (Bruehl et al., 1998).

By contrast to our initial hypotheses, we did not observe differential effects of baroreceptor stimulation on the CPM response between NP and CLBP participants. A significant association was, however, observed between BSI_rest and the magnitude of inhibition/facilitation during CPM in NP and CLBP. Both CLBP and NP participants with lower resting BSI levels exhibited facilitation, comprising an increase in subjective pain perception during CPM. This finding mirrors our previous study, where the CPM response in pain-free individuals was associated with low-frequency heart rate variability (LF-HRV) index (Makovac et al., 2021), a measure associated with baroreflex efficiency (Rahman et al., 2011; Reyes del Paso et al., 2013). Replicating this finding in an independent group of individuals validates our initial hypotheses of an association between the baroreflex and descending pain control capacity. Magnetic resonance imaging studies have shown that brainstem regions important in both invoking CPM responses and elaboration of afferent cardiovascular information, namely the PAG and rostral ventrolateral medulla, are also implicated in the aetiology and maintenance of chronic pain (de Felice et al., 2011; Hemington & Coulombe, 2015). Further magnetic resonance imaging analyses in our previously published study with pain-free participants revealed that the association between LF-HRV and CPM was mediated by the strength of functional connectivity a statistical measure of the association of activity between anatomically distant brain regions) between the periaqueductal grey and the prefrontal cortex (Makovac

et al., 2021). Currently, untangling whether dysregulation of the relationship between autonomic responses and descending pain responses is a result of living with chronic pain or a predisposing factor remains challenging. Longitudinal data are required to provide answers to this question. However, the existence of this association in individuals with no-pain within the present study suggests the presence of a potential vulnerability that may be found in non-clinical populations. We speculate that such vulnerability may contribute to the development of chronic pain and that the successful characterisation of this mechanism may be of benefit for identifying individuals at risk of developing chronic pain.

The present study indicates the potential importance of the baroreflex in pain perception and indicates potential avenues of exploiting these mechanisms to improve treatment of individuals with chronic pain. An improved understanding of the mechanisms underlying these findings may facilitate provision of individualised pain management strategies. Pharmacological or non-pharmacological treatments targeting the baroreflex might also be explored as potential interventions for chronic pain. For example, further exploring the role of the alpha-2 adrenergic system in stimulating baroreflex sensitivity may provide novel pharmacological targets for chronic pain. Although the precise mechanisms remain unknown, dexmedetomidine, an alpha-2 adrenergic receptor agonist with proven baroreflex modulation (Ehara et al., 2012), has demonstrated analgesic properties (Zhao et al., 2020). This observation further reinforces the hypothesis of the central role of the baroreflex in pain modulation. Additionally, non-pharmacological interventions such as slow breathing, stress reduction techniques and heart rate variability biofeedback show promise in alleviating chronic pain (Jafari et al., 2020; Reneau, 2020). Stratifying patients based on underlying pathophysiological mechanisms could help identify those individuals who will more probably benefit from interventions targeting the baroreflex. Ultimately, these advancements have the potential to enhance pain management approaches and improve the quality of life for individuals with chronic pain. Although short-term baroreflex responsiveness to stimulation may exhibit anti-nociceptive effects through increases in BP, this mechanism remains still poorly understood, requiring further research to determine the functional threshold of BP increases for the organism. Indeed, longitudinal studies have revealed that, although the extent of pain inhibition in response to baroreflex stimulation can be beneficial for immediate pain management, it may also predict tonic BP increases after 20 months (Elbert et al., 1994), indicating a trade-off between the costs and benefits of this mechanism in chronic pain conditions.

We acknowledge some methodological limitations in our study. For practical reasons, we collected HR data

using PPG rather than using ECG recordings, the gold standard approach for deriving cardiovascular physiological measures. Some studies have shown that, under certain conditions, PPG can serve as a suitable alternative to ECG (Bolanos et al., 2006; Schäfer & Vagedes, 2013). We are aware of the debates around the optimal CPM paradigm; however, the types of painful stimulation adopted in the present study, namely pressure pain delivered to the thumbnail bed as a test stimulus and cold pain as a conditioning stimulus, have been previously reported as approaches with good reliability (Kennedy et al., 2016). Finally, given the exploratory nature of the study and the overall aim of generating new hypotheses regarding the interaction of baroreceptors in the relationship between the ANS and CPM efficiency, we acknowledge that we applied a more liberal approach by using uncorrected $p$ values. These findings encourage future replications of our results with larger sample sizes.

## Conclusions

Our data provide evidence that indicates the importance of baroreflex functioning in pain modulatory processes in NP and CLBP participants and provide a first experimental attempt to elucidate the role of the baroreceptor in descending pain modulation in individuals with chronic pain. The findings are important for the validation of baroreflex-associated autonomic measures of pain vulnerability and maintenance in chronic pain syndromes. We suggest they provide enticing potential as markers of pain severity and movement towards the much needed mechanism-based profiling of individuals with chronic pain.

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

## Additional information

### Data availability statement

The data that support the findings of this study are available from the corresponding author upon reasonable request.

### Competing interests

The authors declare that they have no competing interests.

### Author contributions

A.V., E.M. and H.F.J. were responsible for data collection. A.V., E.M., M.A.H., H.F.J., M.M. and D.H.S. were responsible for data analysis. A.V., E.M., M.A.H., H.F.J., M.M. and D.H.S. were responsible for manuscript preparation. E.M. and M.A.H. were responsible for the design of the experiment. M.A.H. was responsible for supervision of data collection. All authors edited and revised the manuscript and agree to be accountable for all aspects of the work. All persons designated as authors qualify for authorship and all those who qualify for authorship are listed.

### Funding

This study was funded by the EFIC-Grunenthal Award 2019. MH was supported by the NIHR Biomedical Research Centre and Clinical Research Facility at South London and Maudsley NHS Foundation Trust and King's College London.

### Acknowledgements

We thank all of the volunteers for their participation in the study and the Brixton Therapy Centre for their help with recruitment. We also express our gratitude to Dr Giovanni Calcagnini for his contribution to the development of the baroreceptor stimulating machine and extend our appreciation to Alfonso De Lara-Rubio and Simon Hill for the realisation of the associated hardware and software.

## Keywords

artificial baroreceptor stimulation, baroreceptor sensitivity index, chronic lower back pain, conditioned pain modulation, pressure pain

## Supporting information

Additional supporting information can be found online in the Supporting Information section at the end of the HTML view of the article. Supporting information files available:

**Peer Review History**

