## [Peer Review History · The Journal of Physiology]

Investigating the Effects of Artificial Baroreflex Stimulation on Pain Perception: A Comparative Study in No-pain and Chronic Low Back Pain Individuals

Alessandra Venezia, Harriet Fawsitt-Jones, David Hohenschurz-Schmidt, Matteo Mancini, Matthew A Howard, and Elena Makovac

DOI: 10.1113/JP286375

Corresponding author(s): Alessandra Venezia (alessandra.1.venezia@kcl.ac.uk)

Review Timeline:

Submission Date:	13-Feb-2024
Editorial Decision:	15-Mar-2024
Revision Received:	31-Jul-2024
Editorial Decision:	15-Aug-2024
Revision Received:	30-Aug-2024
Accepted:	02-Sep-2024

Senior Editor: David Wyllie

Reviewing Editor: Vaughan Macefield

Transaction Report:

Dear Dr Makovac,

Re: JP-RP-2024-286375 "Investigating the Effects of Artificial Baroreflex Stimulation on Pain Perception: A Comparative Study in Healthy and Chronic Low Back Pain Individuals" by Alessandra Venezia, Harriet Fawsitt-Jones, David Hohenschurz-Schmidt, Matteo Mancini, Matthew A Howard, and Elena Makovac

Thank you for submitting your manuscript to The Journal of Physiology. It has been assessed by a Reviewing Editor and by 2 expert referees and we are pleased to tell you that it is acceptable for publication following satisfactory revision.

REVISION CHECKLIST:

Please upload two versions of your manuscript text: one with all relevant changes highlighted and one clean version with no changes tracked. The manuscript file should include all tables and figure legends, but each figure/graph should be uploaded as separate, high-resolution files. The journal is now integrated with Wiley's Image Checking service. For further details, see: <https://www.wiley.com/en-us/network/publishing/research-publishing/trending-stories/upholding-image-integrity-wileys-image-screening-service>.

- 'Potential Cover Art' for consideration as the issue's cover image
- Appropriate Supporting Information (video, audio or data set: see https://jp.msubmit.net/cgi-bin/main.plex?form_type=display_requirements#supp)

We look forward to receiving your revised submission.

Yours sincerely,

David Wyllie
Senior Editor
The Journal of Physiology

REQUIRED ITEMS

- Author photo and profile. First or joint first authors are asked to provide a short biography (no more than 100 words for one author or 150 words in total for joint first authors) and a portrait photograph. These should be uploaded and clearly labelled together in a Word document with the revised version of the manuscript. See Information for Authors for further details.
- The reference list must be in alphabetical order, rather than numbered, to comply with our Journal format.
- The Journal of Physiology funds authors of provisionally accepted papers to use the premium BioRender site to create high resolution schematic figures. Follow this link and enter your details and the manuscript number to create and download figures. Upload these as the figure files for your revised submission. If you choose not to take up this offer, we require figures to be of similar quality and resolution. If you are opting out of this service to authors, state this in the Comments section on the Detailed Information page of the submission form. The link provided should only be used for the purposes of this submission. Authors will be charged for figures created on this premium BioRender account if they are not related to this manuscript submission.
- Please upload separate high-quality figure files via the submission form.
- Please ensure that any tables are editable and in Word format, and wherever possible, embedded in the article file itself.
- Please ensure that the Article File you upload is a Word file.
- Papers must comply with the Statistics Policy: https://jp.msubmit.net/cgi-bin/main.plex?form_type=display_requirements#statistics.

In summary:

- If n {less than or equal to} 30, all data points must be plotted in the figure in a way that reveals their range and distribution. A bar graph with data points overlaid, a box and whisker plot or a violin plot (preferably with data points included) are acceptable formats.
- If $n > 30$, then the entire raw dataset must be made available either as supporting information, or hosted on a not-for-profit repository, e.g. FigShare, with access details provided in the manuscript.
- 'n' clearly defined (e.g. x cells from y slices in z animals) in the Methods. Authors should be mindful of pseudoreplication.

- All relevant 'n' values must be clearly stated in the main text, figures and tables.
- The most appropriate summary statistic (e.g. mean or median and standard deviation) must be used. Standard Error of the Mean (SEM) alone is not permitted.
- Exact p values must be stated. Authors must not use 'greater than' or 'less than'. Exact p values must be stated to three significant figures even when 'no statistical significance' is claimed.

- Please include an Abstract Figure file, as well as the Figure Legend text within the main article file. The Abstract Figure is a piece of artwork designed to give readers an immediate understanding of the research and should summarise the main conclusions. If possible, the image should be easily 'readable' from left to right or top to bottom. It should show the physiological relevance of the manuscript so readers can assess the importance and content of its findings. Abstract Figures should not merely recapitulate other figures in the manuscript. Please try to keep the diagram as simple as possible and without superfluous information that may distract from the main conclusion(s). Abstract Figures must be provided by authors no later than the revised manuscript stage and should be uploaded as a separate file during online submission labelled as File Type 'Abstract Figure'. Please also ensure that you include the figure legend in the main article file. All Abstract Figures should be created using BioRender. Authors should use The Journal's premium BioRender account to export high-resolution images. Details on how to use and access the premium account are included as part of this email.

- Please include a full title page as part of your main article (Word) file, which should contain the following: title, authors, affiliations, corresponding author name and contact details, keywords, and running title.

- Please ensure that all figures and tables have a title and legend, and that they have been cited within the main article text.

EDITOR COMMENTS

Reviewing Editor:

Thank you for submitting your manuscript to The Journal of Physiology. I have now received comments from two independent assessors, both experts in the field. As you will see from their comments, they are generally positive about your manuscript but do have a series of recommendations you will need to address, not the least being providing more clarity in the presentation of your methods and results. In particular, as raised by Reviewer 1, it is important to consider inter-individual differences in susceptibility to conditioned pain modulation; indeed, simply pooling the data hides important information on whether chronic pain affects this capacity.

Please add individual data points to the bar histograms of Fig. 7

Please also see 'Required Items' above.

Senior Editor:

Thank you for submitting this work to The Journal of Physiology. Your manuscript has been assessed positively by two expert referees each of whom raise points that I feel you will be able to address in a revised version of your manuscript. Specifically, as highlighted by the Reviewing Editor, there could be a better presentation of some of the methods and results. Please ensure you address all the points raised.

While you report p values our policy is to report to 3 significant figures (unless $p < 0.001$)

REFEREE COMMENTS

Referee #1:

This study aimed to explore the relationship between baroreflex function and pain modulation. The study is well-designed and conducted and the results are extremely interesting. The methods are clearly described and the data obtained is sound. The manuscript is easy to read and well-written, although I have suggested a few statements that could be toned down a little. I list a number of minor comments below that could improve the overall manuscript. I have been picky because I think it's a really nice study and so I was really interested in the results and their context. I suggest one major point below - which I think might result in a different spin on the CPM finding that you have presented.

Major point:

1) An important point to consider. When defining an individual's CPM response, the variability of the baseline pain intensity ratings can be a critical issue. It is known that individuals can display variable pain ratings to the same repeated noxious stimuli (some shown consistent responses to repeated stimuli). It is critical to take this into account where determining an individual CPM response. I appreciate that you have determined whether baroreceptor stimulation has resulted in a change in CPM responsiveness using all participants as a single group. It would be useful to also explore the effects of baroreceptor stimulation only in those individuals that show a significant CPM analgesic response and only in those that do not show a response. Does the proportion of individuals in each group change during baroreceptor stimulation? You can define responders and non-responders using for example permutation testing or a 2SD method (see Crawford et al 2021 and others). This is even more important when looking at a group of chronic pain patients given that there is the possibility that their CPM ability is reduced meaning even greater number are likely to be CPM non-responder.

Minor points:

2) The authors should be a little more careful with some statements in the Introduction.

Sentence 2: I appreciate you referenced a review however there is work by Macefield etc that shows significant variability in the BP, HR and sympathetic responses to acute cutaneous and muscle pain in HC. In addition, there is evidence from experimental animal (Bandler, Carrive etc) and human observations (Lewis in the 1950s for example) that show/observe that pain originating in different tissues likely produces different cardiovascular responses. I would suggest tempering these statements.

3) Page 4, first paragraph: the authors state that patients receiving ACE inhibitors are more likely to have chronic pain than those that are not treated. Is it not the case that those on ACE inhibitors have controlled BP? I think the argument that high blood pressure may protect against chronic pain is extremely tenuous argument. I can see that in individuals with chronic pain, higher BP may be associated with lower perceived pain intensity, however the statement implies that having high blood pressure decreases your propensity for developing chronic pain in the first place. Is there solid evidence to support this?

4) Page 4, paragraph 2, last sentence: please check the reference is correct.

5) Page 4, para 3, sentence 2: Please temper this - the stimulus used in CPM is not always standardized - it can be muscle, pressure, heat etc and I do not think there is evidence that the BP effects of these stimuli are consistent between stimuli and in particular between individuals.

6) Page 5, para 1, sentence 3: Again, please rework - you state earlier that not all studies shown that chronic pain is associated with reduced CPM ability - i.e. back pain - thus your statement here appears to be definitive, yet CPM changes are not.

7) Page 7, last paragraph. I appreciate that the authors measured state and trait pain. However, it has also been shown that in many individuals with chronic pain, the intensity of pain can fluctuate wildly over even a short period of time whereas in others it is stable (see Mills et al) - thus I would recommend in future studies measure an individual on-going pain at regular intervals during the entire experimental procedure to get a better understanding of the individual pain intensity profile.

8) Were the correlations corrected for multiple comparisons?

9) Please add to Figure legend 7 what the plots are - mean? + STDEV or SEM?? Also, I assume for each individual the pain ratings were averaged across all the relevant trials so that one value per participant contributes to these plots? This should be stated in the Methods.

10) It might be useful to add plots of potential the changes in CPM -related VAS scores during active versus sham

baroreceptor stimulation? Maybe you could combine them into one figure 7? I realize there was no significant effect, but it would nicely contrast the results you see with pressure pain stimuli alone.

11) Figure 8: please explain what the CPM values on the x-axis represent. Do more negative values represent greater reductions in pain during dual versus single noxious stimuli - i.e. greater CPM analgesic response?

12) This plot also reinforces my opening comment with respect to exploring individuals in responder and non-responder groups. These plots look like during active stimulation more participants are at z-axis "0" (I assume no CPM response) compared with during sham baroreceptor stimulation.

13) Page 25, para 2, sentence 1: Again, I would rework the idea that high BP is "protective" against pain. It might be associated with pain intensity but does not necessarily "protect" against it.

Referee #2:

Interesting study that would benefit from some greater clarity / specificity in the writing: Viz.

One of the key points should be perhaps an implication for pain management

General/abstract

Important underexplored area of study.

In the abstract, could be more specific about 'relationship between baroreflex functioning..' is this baseline BRS, or similar, or the (baroreflex) response to baroreceptor stimulation that modulated pressure pain? IE was the baroreflex per se involved in the group difference in pain perception ? I see the CPM scores did show baseline relationship with BRE though, so may be order needs considering in tightening clarity of abstract.

Introduction: I am a fan of termed 'healthy controls', even when people are screening in depth for physical and psychological ailments. 'Controls' typically refers to members of comparison group - hence inappropriate term for studies when there is no comparison between groups.

Would be better to explain clearly the CPM paradigm and associated measurement than just say it is widely used and that it is inefficient in some patients. Also similarly if including baroreflex efficiency within a hypothesis , it would be useful to say what the baroreflex is and how efficiency is measured (and relates to other measures of baroreflex function).

Method: Unclear what the difference between the pulse oximetry and caretaker measurement are - is it an algorithm relating to stroke /pulse volume in Caretaker - was this then the basis of deriv3ed blood pressure measure? Was this calibrated? Calculation of baroreflex Sensitivity relies on this e.g. slowing on next pulse to preceding high stroke volume / BP? The approach used for calculating baroreflex efficiency (referred to in abstract) is not described in methods. Were group sizes calculated a priori? Sample size is relatively small for correlational analyses. There are a number of contrasts and correlations tested - were any corrections applied for multiple comparisons?

Results: Table 2 perhaps contradicts the claim that the group was fully age matched.

I am unclear why BSI_stress is used rather than a difference between (BSI_stress - BSI_rest) to examine effect of stress on pain etc for other measures e.g. HR.

I am unclear if baroreceptor stimulation affected autonomic measures other than pulse interval. If not, how was it validated that the stimulation stimulated baroreceptors? Also were participants blind to stimulation and sham conditions?

The results do not report measures of baroreflex efficiency referred to in abstract

Discussion: Interesting findings were observed including a difference from prior studies regarding relationship between chronic pain, pain perception and BP. I am unclear from the results to what extent the discussion can refer to activation of the baroreflex, as differential effects of stimulation / cold on BSI or HR aren't presented clearly in the results section.

The discussion should link inferences and past literature more closely to the specific findings in this study.

END OF COMMENTS

Confidential Review

13-Feb-2024

Dear Editor and Reviewers,

First, we would like to thank you for taking the time to provide in-depth analysis of our study followed by valuable and constructive criticisms. We also thank the Reviewers in acknowledging the potential of our findings. The criticisms raised by the two Reviewers and the Editor are valid and we have provided additional analyses to allay methodological concerns.

We are confident that the Reviewers will agree that our efforts have led to a much stronger manuscript. We are convinced that results are interesting and are informative to other scientists interested in brain-body interactions sub-serving the perception and regulation of pain. All the modifications have been highlighted in yellow in the revised manuscript.

Additional minor changes have been implemented as consequence of proof reading and revising:

- RR_active has been changed to HR_active in section 2.5.4.
- We have realized that we did not use age as a covariate in our original ANOVA on effect of baroreflex stimulation on pressure pain (section 2.5.5. Analysis of behavioural data from PRESSURE PAIN paradigm). We have re-run the analysis and updated the values on the ANOVA. The pattern of results remains unchanged.
- We have realized a mistake in the calculations of the kPa used to induce pressure pain and this has been corrected in the Method section.
- Section 3.7, we clarify that due to artefacts in the pulseoximeter, analysis was performed in 20 patients and 24 HC.
- Editing changes: RR substituted for inter-beat-interval to facilitate reading; the term pulseoximeter changed for

EDITOR COMMENTS

Reviewing Editor:

Reviewing Editor Comment #1: Thank you for submitting your manuscript to The Journal of Physiology. I have now received comments from two independent assessors, both experts in the field. As you will see from their comments, they are generally positive about your manuscript but do have a series of recommendations you will need to address, not the least being providing more clarity in the presentation of your methods and results. In particular, as raised by Reviewer 1, it is important to consider inter-individual differences in susceptibility to conditioned pain modulation; indeed, simply pooling the data hides important information on whether chronic pain affects this capacity.

Reviewing Editor Comment #2. Please add individual data points to the bar histograms of Fig. 7

Response: Individual data points have been added to Figure 7 as requested.

Reviewing Editor Comment #3. Please also see 'Required Items' above.

Senior Editor:

Senior Editor Comment #1: Thank you for submitting this work to The Journal of Physiology. Your manuscript has been assessed positively by two expert referees each of whom raise points that I feel you will be able to address in a revised version of your manuscript. Specifically, as highlighted by the Reviewing Editor, there could be a better presentation of some of the methods and results. Please ensure you address all the points raised.

Senior Editor Comment #2: While you report p values our policy is to report to 3 significant figures (unless $p < 0.001$)

Response: This has been updated as per policy.

REFEREE COMMENTS

Referee #1:

This study aimed to explore the relationship between baroreflex function and pain modulation. The study is well-designed and conducted and the results are extremely interesting. The methods are clearly described, and the data obtained is sound. The manuscript is easy to read and well-written, although I have suggested a few statements that could be toned down a little. I list a number of minor comments below that could improve the overall manuscript. I have been picky because I think it's a really nice study and so I was really interested in the results and their context. I suggest one major point below - which I think might result in a different spin on the CPM finding that you have presented.

Referee #1 Comment #1: An important point to consider. When defining an individual's CPM response, the variability of the baseline pain intensity ratings can be a critical issue. It is known that individuals can display variable pain ratings to the same repeated noxious stimuli (some shown consistent responses to repeated stimuli). It is critical to take this into account when determining an individual CPM response. I appreciate that you have determined whether baroreceptor stimulation has resulted in a change in CPM responsiveness using all participants as a single group. It would be useful to also explore the effects of baroreceptor stimulation only in those individuals that show a significant CPM analgesic response and only in those that do not show a response. Does the proportion of individuals in each group change during baroreceptor stimulation? You can define responders and non-responders using for example permutation testing or a 2SD method (see Crawford et al 2021 and others). This is even more important when looking at a group of chronic pain patients given that there is the possibility that their CPM ability is reduced meaning even greater numbers are likely to be CPM non-responders.

Response: We would like to thank the reviewer for this interesting suggestion. We are indeed aware that baseline variability is an important factor to consider, yet it is not easily implemented in our study due to the limited number of trials. In fact, the CPM session consisted of only 2 pressure trials and 2 CPM trials. The mean values of the 2 pressure-only trials were highly correlated with the mean value of the 'Pressure-only' experiment (which has 16 pressure-only trials). The choice of having only a limited number of trials was based on our previous extensive piloting with the same pressure + CPM paradigm, which showed a sensitization effect that in some cases overrode the CPM effect (i.e., in the last trials of the session, we observed a temporal summation effect on perceived pain). This prompted us to reduce the number of trials in accordance with other similar studies using CPM.

This interesting suggestion put forward by the reviewer is certainly one that we will keep in mind in our future studies. Indeed, we are currently trialling different CPM paradigms, including the use of a pressure-algometer, which show a more robust CPM (inhibitory) and a controlled pain sensitization effect.

At present, our correlational results allow us to further disentangle some factors that likely contribute to the well-documented inter-individual variability of the CPM response. Our data add to this knowledge by showing that the reactivity of the baroreflex, which is likely associated with other variables such as chronic stress response and emotional regulation, can explain a portion of this variability.

Referee #1 Comment #2: The authors should be a little more careful with some statements in the Introduction. Sentence 2: I appreciate you referenced a review however there is work by Macefield et al that shows significant variability in the BP, HR and sympathetic responses to acute cutaneous and muscle pain in HC. In addition, there is evidence from experimental animal (Bandler, Carrive et al) and human observations

(Lewis in the 1950s for example) that show/observe that pain originating in different tissues likely produces different cardiovascular responses. I would suggest tempering these statements.

Response: We have added the relevant reference to our Introduction. We are now differentiating the effect of acute pain from the reported effect of long-lasting tonic pain originating in deep tissues as described in Macfield and colleagues. Thank you for pointing out to this incongruence.

Referee #1 Comment #3: Page 4, first paragraph: the authors state that patients receiving ACE inhibitors are more likely to have chronic pain than those that are not treated. Is it not the case that those on ACE inhibitors have controlled BP? I think the argument that high blood pressure may protect against chronic pain is extremely tenuous argument. I can see that in individuals with chronic pain, higher BP may be associated with lower perceived pain intensity, however the statement implies that having high blood pressure decreases you propensity for developing chronic pain in the first place. Is there solid evidence to support this?

Response: We agree with the reviewer that the statement on ACE inhibitors might be unsupported by the current evidence, and we have therefore removed the sentence and the reference.

Additionally, we acknowledge that our description of the complex (and still not completely understood) relationship between BP and pain lacked sufficient detail and might have led the reader to misunderstand that a condition of clinically high blood pressure might be protective against pain. This is indeed incorrect, as many studies indicate a higher prevalence of hypertension in people with persistent pain (e.g., Parsons et al., 2015).

On the other side, in patients with coronary diseases, an inverse relation between chest pain and BP both at rest (Falcone et al, 1997, Ditto et al.,2010) and during physical activity (Ditto et al., 2007) has been documented. Along this lines, Makovac' meta-analysis confirms an association between BP and pain, both with psychological and physiological measures, and other studies unequivocally point to an association between baroreflex and pain (Makovac et al., 2020. doi: 10.1097/HJH.0000000000002427.), likely being the mechanism that explains the better-investigated condition of stress-analgesia. One possible explanation, which requires further investigation, is that a temporary increase in BP, such as that experienced during a fight-or-flight response, is pain protective and has a clear evolutionary advantage. This association can still be observed in people with hypertension. However, a persistent condition of elevated BP is not functional. For instance, a blunted or dysregulated baroreflex due to chronic stress might lose its inhibiting influence on pain. The issue of causality remains unresolved and will need further consideration in future studies. For example, it is psychological stress that acts as a factor in the dysregulation of the baroreflex, subsequently increasing the likelihood of developing a chronic pain condition. Or, alternatively, chronic pain (being a continuous physiological stressor to the organism) lead to this dysregulation.

Whether efficient baroreflex regulation is associated with a lower risk of developing chronic pain is not well understood. Studies seem to suggest that reduced heart rate variability and baroreflex sensitivity are known markers for hypertension risk, whilst it is currently unknown whether the opposite is true as well. Acknowledging the lack of clarity on this point, we have added new information in the first paragraph of our introduction, which will hopefully provide more clarity on the suggested mechanisms but also highlight the current incongruence in the literature and the knowledge gaps.

Referee #1 Comment #4: Page 4, paragraph 2, last sentence: please check the reference is correct.

Response: Thank you for spotting this mistake. We have updated the correct reference:

Ocay DD, Ye D-L, Larche CL, et al.: Clusters of facilitatory and inhibitory conditioned pain modulation responses in a large sample of children, adolescents, and young adults with chronic pain. PAIN Reports. 2022, 7:e1032.

Referee #1 Comment #5: Page 4, para 3, sentence 2: Please temper this - the stimulus used in CPM is not always standardized - it can be muscle, pressure, heat etc and I do not think there is evidence that the BP effects of these stimuli are consistent between stimuli and in particular between individuals.

Response: The sentence has been changed. Indeed, we agree that the substantial inter-individual variability in response to a painful stimulation might explain some of the CPM variability. This has also been added as a potential mechanisms.

Referee #1 Comment #6: Page 5, para 1, sentence 3: Again, please rework - you state earlier that not all studies shown that chronic pain is associated with reduced CPM ability - i.e. back pain - thus your statement here appears to be definitive, yet CPM changes are not.

Response: We have revised the sentence to emphasize that the extent of autonomic dysregulation could be a factor contributing to the variability of the CPM response in chronic pain conditions.

Referee #1 Comment #7: Page 7, last paragraph. I appreciate that the authors measured state and trait pain. However, it has also been shown that in many individuals with chronic pain, the intensity of pain can fluctuate wildly over even a short period of time whereas in others it is stable (see Mills et al) - thus I would recommend in future studies measure an individual on-going pain at regular intervals during the entire experimental procedure to get a better understanding of the individual pain intensity profile.

Response: We agree with the Reviewers comment, and we will indeed keep this in mind in our current ongoing and future studies.

Referee #1 Comment #8: Were the correlations corrected for multiple comparisons?

Response: Given the exploratory nature of our study, we did not to apply multiple corrections for p-values. Our approach was driven by a concern that applying such corrections, while useful in reducing the risk of Type I errors (false positives), can substantially increase the risk of Type II errors (false negatives), potentially obscuring meaningful associations that warrant further investigation. The primary goal of this research was to generate hypotheses and identify potential areas for further study rather than to confirm definitive relationships. We hope that the reviewer will agree that the strong nature of our hypotheses coupled by the highly innovative and explorative approach justifies a more liberal approach to multiple comparisons. We are now highlighting this in our discussion, and call for future replications of our results with larger sample sizes.

Referee #1 Comment #9: Please add to Figure legend 7 what the plots are - mean? + STDEV or SEM?? Also, I assume for each individual the pain ratings were averaged across all the relevant trials so that one value per participant contributes to these plots? This should be stated in the Methods.

Response: We have included a description of the plots, which represent the mean \pm SEM. We have averaged all the values across the relevant trials (pressure sham and pressure active) for each participant, as clarified in the Methods section.

Referee #1 Comment #10: It might be useful to add plots of potential the changes in CPM -related VAS scores during active versus sham baroreceptor stimulation? Maybe you could combine them into one figure 7? I realize there was no significant effect, but it would nicely contrast the results you see with pressure pain stimuli alone.

Response: We have updated Figure 7 as requested by the reviewer.

Referee #1 Comment #11: Figure 8: please explain what the CPM values on the x-axis represent. Do more negative values represent greater reductions in pain during dual versus single noxious stimuli - i.e. greater CPM analgesic response?

Response: The direction of the delta values has been described in the Method section, where we state that: ' differential CPM scores were calculated separately for the ACTIVE and SHAM conditions, with the following formula: $CPM_{active/sham} = VAS ('CPM') - VAS ('Pressure\ only')$ (Table 1). Here, negative values indicate a decrease in perceived pressure pain during the CPM condition, whereas positive numbers indicate an increase in perceived pressure pain.' This has now been described also under Figure 8 to facilitate interpretation.

Referee #1 Comment #12: This plot also reinforces my opening comment with respect to exploring individuals in responder and non-responder groups. These plots look like during active stimulation more participants are at z-axis "0" (I assume no CPM response) compared with during sham baroreceptor stimulation.

Response: Please refer to our response to Comment #1 where we explain why, on this occasion, the division of our participants into CPM responders and non-responders is not possible. We agree that our data might potentially reveal more results, yet unfortunately this study is not adequately powered to explore these nuances. We would also tentatively speculate that, given our primary manipulation is baroreflex stimulation, a first division to explore is whether BRS responders and non-responders have a different CPM pattern, which is certainly something we will keep in mind in our future, fully powered studies.

Referee #1 Comment #13: Page 25, para 2, sentence 1: Again, I would rework the idea that high BP is "protective" against pain. It might be associated with pain intensity but does not necessarily "protect" against it.

Response: Please refer to Comment 3 where we address this criticism.

Referee #2

Referee #2 Comment #1: One of the key points should be perhaps an implication for pain management.

Response: We kindly direct the reviewer's attention to our discussion paragraph where we enumerate the potential implications of our data for pain management:

This study indicates the potential importance of the baroreflex in pain perception and indicates potential avenues of exploiting these mechanisms to improve treatment of patients with chronic pain. Improved understanding of the mechanisms underlying these findings may facilitate provision of individualised pain management strategies. Pharmacological or non-pharmacological treatments targeting the baroreflex might also be explored as potential interventions for chronic pain. For example, further exploring the role of the alpha-2 adrenergic system in stimulating baroreflex sensitivity may provide to novel pharmacological targets for chronic pain. While the precise mechanisms remain unknown, dexmedetomidine, an α 2-adrenergic receptor agonist with proven baroreflex modulation (47), has demonstrated analgesic properties (48). This observation further reinforces the hypothesis of the central role of the baroreflex in pain modulation. Additionally, non-pharmacological interventions such as slow breathing, stress reduction techniques, and heart rate variability (HRV) biofeedback show promise in alleviating chronic pain (49, 50). Stratifying patients based on underlying pathophysiological mechanisms could help identify those who are more likely to benefit from interventions targeting the baroreflex. Ultimately, these advancements have the potential to enhance pain management approaches and improve the quality of life for individuals with chronic pain.

Referee #2 Comment #2: In the abstract, could be more specific about 'relationship between baroreflex functioning.' is this baseline BRS, or similar, or the (baroreflex) response to baroreceptor stimulation that modulated pressure pain? IE was the baroreflex per se involved in the group difference in pain perception? I see the CPM scores did show baseline relationship with BRE though, so may be order needs considering in tightening clarity of abstract.

Response: Thank you for providing us with the opportunity to clarify this point, which we indeed agree might be confusing to the reader. We have reworded the sentence to clarify that there was a differential modulation of our stimulation on pain perception. Please refer to the highlighted sentence in the manuscript.

Referee #2 Comment #3: Introduction: I am a fan of termed 'healthy controls', even when people are screening in depth for physical and psychological ailments. 'Controls' typically refers to members of comparison group - hence inappropriate term for studies when there is no comparison between groups.

Response: We apologise, but we are unsure whether we fully understand the Referee's comment. Specifically, we are uncertain whether the Referee is suggesting using the term "Controls" instead of "Healthy controls." We understand that the term "healthy" might be inappropriate and overly defining, especially in conditions where the participants have not undergone a comprehensive medical screening before the experiment. We can, however, assure that our eligibility criteria were very stringent, not allowing for the participation of any individuals with diagnosed clinical conditions. Additionally, participants underwent an autonomic assessment, ruling out autonomic conditions. With the exception of the rare circumstance of unknown underlying medical conditions at the time of the assessment and intentional deception, we are confident that our control group was in a clinically healthy status at the time of testing.

Referee #2 Comment #4: Would be better to explain clearly the CPM paradigm and associated measurement than just say it is widely used and that it is inefficient in some patients. Also similarly if including baroreflex efficiency within a hypothesis, it would be useful to say what the baroreflex is and how efficiency is measured (and relates to other measures of baroreflex function).

Response: As recommended, a definition and operationalisation of CPM and baroreflex efficiency have been incorporated into the Introduction.

' Conditioned Pain Modulation (CPM) is a widely used paradigm for assessing descending pain modulation in humans, typically performed by applying a painful conditioning stimulus to one part of the body to reduce the pain perception from a test stimulus applied to another part of the body.'

' This relationship might be mediated by the baroreceptors, specialized sensory receptors located in the walls of the carotid sinus, which regulate cardiovascular functions by adjusting heart rate and blood pressure to maintain homeostasis (6, 7). Indeed, heightened baroreflex sensitivity, assessed by the rate of heart rate response to fluctuations in blood pressure (7), has been found to correlate inversely with the severity of pain experienced by healthy individuals (8) and can predict post-surgical pain ((9).'

The following citations have been added to support the new claims in the introduction:

- Makovac E, Porciello G, Palomba D, Basile B, Ottaviani C: Blood pressure-related hypoalgesia: a systematic review and meta-analysis. *Journal of Hypertension*. 2020, 38:1420-1435.
- Swenne C: Baroreflex sensitivity: mechanisms and measurement. *Netherlands Heart Journal*. 2013, 21:58-- - Duschek S, Mück I, Del Paso GR: Relationship between baroreceptor cardiac reflex sensitivity and pain experience in normotensive individuals. *International Journal of Psychophysiology*. 2007, 65:193-200.
- Suarez-Roca H, Mamoun N, Watkins LL, Bortsov AV, Mathew JP: Higher Cardiovascular Baroreflex Sensitivity Predicts Increased Pain Outcomes After Cardiothoracic Surgery. *The Journal of Pain*. 2024, 25:187-201.

Referee #2 Comment #5: Method: Unclear what the difference between the pulse oximetry and caretaker measurement are - is it an algorithm relating to stroke /pulse volume in Caretaker - was this then the basis of derived blood pressure measure? Was this calibrated? Calculation of baroreflex Sensitivity relies on this e.g. slowing on next pulse to preceding high stroke volume / BP? The approach used for calculating baroreflex efficiency (referred to in abstract) is not described in methods.

Response: Thank you for highlighting the lack of clarity regarding this point. The key difference between the two devices is that the CareTaker allows for simultaneous acquisition of both blood pressure BP and heart rate HR data, whereas a traditional pulse oximeter only collects HR data. The CareTaker Blood Pressure Monitor uses a finger sensor with photoplethysmography (PPG) technology to detect blood volume changes in the microvascular bed of tissue. Algorithms analyse these changes to estimate systolic and diastolic BP. The device is calibrated to each individual's baseline BP. This detail has now been added to the Methods section.

Further to the point about baroreflex efficiency, the referee correctly identified the lack of description of this index. In our manuscript, we have described and used the baroreflex sensitivity (BRS) index in subsequent analyses. Mentioning baroreflex efficiency, which we acknowledge as a different measure, was a mistake on our part. We have now corrected this, and we explicitly refer to baroreflex sensitivity in the abstract and throughout the manuscript.

Referee #2 Comment #6: Were group sizes calculated a priori? Sample size is relatively small for correlational analyses. There are a number of contrasts and correlations tested - were any corrections applied for multiple comparisons?

Response: Given the exploratory nature of our study, we did not to apply multiple corrections for p-values. Our approach was driven by a concern that applying such corrections, while useful in reducing the risk of Type I errors (false positives), can substantially increase the risk of Type II errors (false negatives), potentially obscuring meaningful associations that warrant further investigation. The primary goal of this research was to generate hypotheses and identify potential areas for further study rather than to confirm definitive relationships. We hope that the reviewer will agree that the strong nature of our hypotheses coupled by the highly innovative and explorative approach justifies a more liberal approach to multiple comparisons. We are now highlighting this in our discussion and call for future replications of our results with larger sample sizes. Similarly, a priori group sizes were not calculated, but instead sample size was based on previous studies conducted in our group with CPM and our previous studies with baroreceptor stimulation. Data from this study are used for sample size calculations of our ongoing studies.

Referee #2 Comment #7: Results: Table 2 perhaps contradicts the claim that the group was fully age matched.

Response: Indeed, despite not meeting the arbitrary threshold of statistical significance ($p=005$), there was trend towards our HC being younger as compared to CLBP patients. This has now been explicitly mentioned in the Result section. To address this point, age and gender were included as covariates in all analyses.

Referee #2 Comment #8: I am unclear why BSI_stress is used rather than a difference between (BSI_stress - BSI_rest) to examine effect of stress on pain etc for other measures e.g. HR.

Response: We used both the difference and the values of BSI at rest and during stress. While we noticed a consistent trend in the same direction with the difference score, it did not reach statistical significance. This suggests there may be a non-linear relationship between the relative change and pain, which was better captured by the BSI values during stress.

Referee #2 Comment #9: I am unclear if baroreceptor stimulation affected autonomic measures other than pulse interval. If not, how was it validated that the stimulation stimulated baroreceptors? Also were participants blind to stimulation and sham conditions?

Response: Our baroreceptor stimulating machine has been validated in multiple previous studies, either by our collaborators (Mancini et al., 2014; Calcagnini et al., 2001, Basile et al., 2013), or in previous publications conducted by Dr. Makovac (Makovac et al., 2018; Makovac et al 2015), and hence the confidence in the mechanism underlying the stimulation supported by a reliable effect observed in these studies.

Indeed, previous studies have also highlighted that a proportion of participants did not show a significant modulation of the physiology (as explored by a paired ttest comparison the sham to the stimulation in each individual), and were treated as non-responders. In some publications have eliminated the baroreceptor stimulation non-responders from the analyses (i.e., participants who did not show a statistically significant modulation of physiological parameters). In other studies, we have included both responders and non-responders assuming that the degree of the effect is distributed on a continuum and still meaningful (REF). Finally, a differential effect was observed in previous studies depending on the experimental condition (i.e., observable with only fearful and not neutral faces, and only in the ACTIVE and not SHAM condition), indicating that our effect is a genuine result of the stimulation. Finally, participants in this and previous studies

were debriefed at the end of the experiment and did not report to have noticed any difference between the cuff-pressure stimulations.

We respectfully disagree with the Referees that the pulse interval is not a valid measure used as a basis for demonstrating baroreceptor stimulation validity as it is the primary function modulated by the baroreflex in response to changing BP. Other than R-R interval, a more direct measure of the efficiency of our stimulation would be the direct electrical measurement from the baroreceptors, which is understandably impossible to achieve under these experimental circumstances.

Finally, we confirm that the participants were blinded to stimulation and sham conditions, and this has been clarified in the Method section.

References:

Mancini, Matteo, et al. "Modeling heart beat dynamics and fMRI signals during carotid stimulation by neck suction." *2014 36th Annual International Conference of the IEEE Engineering in Medicine and Biology Society*. IEEE, 2014.

Calcagnini, G., et al. "Baroreceptor-sensitive fluctuations of heart rate and pupil diameter." *2001 Conference Proceedings of the 23rd Annual International Conference of the IEEE Engineering in Medicine and Biology Society*. Vol. 1. IEEE, 2001.

Makovac, Elena, et al. "Fear processing is differentially affected by lateralized stimulation of carotid baroreceptors." *Cortex* 99 (2018): 200-212.

Makovac, Elena, et al. "Effect of parasympathetic stimulation on brain activity during appraisal of fearful expressions." *Neuropsychopharmacology* 40.7 (2015): 1649-1658.

Basile, Barbara, et al. "Direct stimulation of the autonomic nervous system modulates activity of the brain at rest and when engaged in a cognitive task." *Human brain mapping* 34.7 (2013): 1605-1614.

Referee #2 Comment #10: The results do not report measures of baroreflex efficiency referred to in abstract.

Response: Thank you for pointing out this incongruity. Indeed, we based our analyses on the Baroreflex Sensitivity index (BSI), which is a different measure from Baroreflex Efficiency. This has been corrected in the Abstract and throughout the manuscript. See also the response to Comment 5 related to this point.

Referee #2 Comment #11: Discussion: Interesting findings were observed including a difference from prior studies regarding relationship between chronic pain, pain perception and BP. I am unclear from the results to what extent the discussion can refer to activation of the baroreflex, as differential effects of stimulation / cold on BSI or HR aren't presented clearly in the results section.

Response: We have carefully reviewed our manuscript in light of these criticisms, aiming to enhance clarity by implementing the following changes:

1. We have reorganized parts of our results for clarity. Specifically, we now sequentially report the effects of autonomic stress on ANS, the effects of baroreceptor stimulation on ANS, the effects of baroreceptor stimulation on pressure pain, and the effects of baroreceptor stimulation on CPM, closely aligning with our primary hypotheses, and hoping that this will facilitate the reader.
2. In the discussion section, we reiterate our primary findings at the outset to assist the reader and reinforce our main conclusions.
3. Although we do not observe a modulating effect of baroreceptor stimulation on Conditioned Pain Modulation, we do find a correlation between the magnitude of CPM modulation in response to baroreceptor stimulation and BSI at rest. This suggests that the responsiveness of the baroreflex may account for some of the inter-individual variability in CPM.

4. Moreover, we do find a differential modulation of BRS on pressure pain in the two groups of patients and HC. We have provided more extensive explanation of this discrepancy in the discussion.

Referee #2 Comment #12: The discussion should link inferences and past literature more closely to the specific findings in this study.

Response: We have added relevant literature and expanded our interpretation of the results. The following literature has been mentioned in the Discussion:

- Bruehl S, Burns JW, McCubbin JA: Altered cardiovascular/pain regulatory relationships in chronic pain. *International Journal of Behavioral Medicine*. 1998, 5:63-75.
- Pflieger M, Straneva PA, Fillingim RB, Maixner W, Girdler SS: Menstrual cycle, blood pressure and ischemic pain sensitivity in women: a preliminary investigation. *International Journal of Psychophysiology*. 1997, 27:161-166.
- Elbert T, Dworkin BR, Rau H, et al.: Sensory effects of baroreceptor activation and perceived stress together predict long-term blood pressure elevations. *International Journal of Behavioral Medicine*. 1994, 1:215-228.
- Fillingim RB, Maixner W, Bunting S, Silva S: Resting blood pressure and thermal pain responses among females: effects on pain unpleasantness but not pain intensity. *International Journal of Psychophysiology*. 1998, 30:313-318.

REQUIRED ITEMS

- (1) Author photo and profile. First or joint first authors are asked to provide a short biography (no more than 100 words for one author or 150 words in total for joint first authors) and a portrait photograph. These should be uploaded and clearly labelled together in a Word document with the revised version of the manuscript. See Information for Authors for further details.
- (2) The reference list must be in alphabetical order, rather than numbered, to comply with our Journal format.
- (3) The Journal of Physiology funds authors of provisionally accepted papers to use the premium BioRender site to create high resolution schematic figures. Follow this link and enter your details and the manuscript number to create and download figures. Upload these as the figure files for your revised submission. If you choose not to take up this offer, we require figures to be of similar quality and resolution. If you are opting out of this service to authors, state this in the Comments section on the Detailed Information page of the submission form. The link provided should only be used for the purposes of this submission. Authors will be charged for figures created on this premium BioRender account if they are not related to this manuscript submission.
- (4) Please upload separate high-quality figure files via the submission form.
- (5) Please ensure that any tables are editable and in Word format, and wherever possible, embedded in the article file itself.
- (6) Please ensure that the Article File you upload is a Word file.

(7) Papers must comply with the Statistics Policy: https://jp.msubmit.net/cgi-bin/main.plex?form_type=display_requirements#statistics.

In summary:

- If n {less than or equal to} 30, all data points must be plotted in the figure in a way that reveals their range and distribution. A bar graph with data points overlaid, a box and whisker plot or a violin plot (preferably with data points included) are acceptable formats.
- If $n > 30$, then the entire raw dataset must be made available either as supporting information, or hosted on a not-for-profit repository, e.g. FigShare, with access details provided in the manuscript.
- 'n' clearly defined (e.g. x cells from y slices in z animals) in the Methods. Authors should be mindful of pseudoreplication.
- All relevant 'n' values must be clearly stated in the main text, figures and tables.
 - The most appropriate summary statistic (e.g. mean or median and standard deviation) must be used. Standard Error of the Mean (SEM) alone is not permitted.
 - Exact p values must be stated. Authors must not use 'greater than' or 'less than'. Exact p values must be stated to three significant figures even when 'no statistical significance' is claimed.
- Please include an Abstract Figure file, as well as the Figure Legend text within the main article file. The Abstract Figure is a piece of artwork designed to give readers an immediate understanding of the research and should summarise the main conclusions.
- If possible, the image should be easily 'readable' from left to right or top to bottom. It should show the physiological relevance of the manuscript so readers can assess the importance and content of its findings. Abstract Figures should not merely recapitulate other figures in the manuscript. Please try to keep the diagram as simple as possible and without superfluous information that may distract from the main conclusion(s). Abstract Figures must be provided by authors no later than the revised manuscript stage and should be uploaded as a separate file during online submission labelled as File Type 'Abstract Figure'. Please also ensure that you include the figure legend in the main article file. All Abstract Figures should be created using BioRender. Authors should use The Journal's premium BioRender account to export high-resolution images. Details on how to use and access the premium account are included as part of this email.
- Please include a full title page as part of your main article (Word) file, which should contain the following: title, authors, affiliations, corresponding author name and contact details, keywords, and running title.
- Please ensure that all figures and tables have a title and legend, and that they have been cited within the main article text.

Dear Dr Venezia,

Re: JP-RP-2024-286375R1 "Investigating the Effects of Artificial Baroreflex Stimulation on Pain Perception: A Comparative Study in Healthy and Chronic Low Back Pain Individuals" by Alessandra Venezia, Harriet Fawsitt-Jones, David Hohenschurz-Schmidt, Matteo Mancini, Matthew A Howard, and Elena Makovac

Thank you for submitting your manuscript to The Journal of Physiology. It has been assessed by a Reviewing Editor and by 2 expert referees and we are pleased to tell you that it is acceptable for publication following satisfactory minor revision.

REVISION CHECKLIST:

- 'Potential Cover Art' for consideration as the issue's cover image

- Appropriate Supporting Information (Video, audio or data set: see https://jp.msubmit.net/cgi-bin/main.plex?form_type=display_requirements#supp).

We look forward to receiving your revised submission.

Yours sincerely,

David Wyllie
Senior Editor
The Journal of Physiology

EDITOR COMMENTS

Reviewing Editor:

Thank you for submitting your revised manuscript to The Journal of Physiology. Both reviewers largely satisfied with your amendments, although Reviewer 2 has some issues remaining regarding the definition of "healthy controls." I agree, and I would like you to define the two groups of participants as those with "no-pain" and those with "chronic pain." Moreover, unless you are seeing those with chronic low-back pain clinically, please do not refer to them as patients but as participants. Please also include details on how these individuals were recruited into the study: was it through advertising within the hospital environment, for example?

Senior Editor:

Thank you for revising your manuscript based on the original reviews. As you will read both referees are happy with the changes/clarifications you have incorporated apart from one point. Referee 2 has issue with your description of "health controls". The Reviewing Editor has suggested alternative wording that I feel could be adopted. With this change, your manuscript will be accepted.

REFEREE COMMENTS

Referee #1:

I thank the authors for their detailed responses to my previous suggestions. All of my suggestions have been addressed and I have no further questions.

Referee #2:

Thank you. The revised manuscript has addressed almost all my earlier concerns for publication. I do want to restate, however, the point that I am NOT a fan of the term 'health controls' as this is presumptive and people in the 'matched non-pain comparison group' were not comprehensively screened to exclude physical or psychological pathology. The sample seemingly had almost half above a 'healthy' BMI and perhaps one or two approached borderline 'unhealthy' alcohol intake.

END OF COMMENTS

Dear Editor and Reviewers,

We would like to thank you again for taking the time to review the manuscript and suggest small but important changes to improve the quality of the paper. All the modifications have been highlighted in yellow in the latest version of the revised manuscript uploaded.

EDITOR COMMENTS

Reviewing Editor Comment 1:

Thank you for submitting your revised manuscript to The Journal of Physiology. Both reviewers largely satisfied with your amendments, although Reviewer 2 has some issues remaining regarding the definition of "healthy controls." I agree, and I would like you to define the two groups of participants as those with "no-pain" and those with "chronic pain." Moreover, unless you are seeing those with chronic low-back pain clinically, please do not refer to them as patients but as participants. Please also include details on how these individuals were recruited into the study: was it through advertising within the hospital environment, for example?

Senior Editor Comment 1:

Thank you for revising your manuscript based on the original reviews. As you will read both referees are happy with the changes/clarifications you have incorporated apart from one point. Referee 2 has issue with your description of "health controls". The Reviewing Editor has suggested alternative wording that I feel could be adopted. With this change, your manuscript will be accepted.

Response: Thank you both for your comments and suggestions. We have replaced the term "healthy controls" with "no-pain participants" and we are now referring to individuals with chronic low back pain as 'participants'. We also specified how participants were recruited in the 2.2 paragraph.

REFEREE COMMENTS

Referee #1:

I thank the authors for their detailed responses to my previous suggestions. All of my suggestions have been addressed and I have no further questions.

Referee #2:

Thank you. The revised manuscript has addressed almost all my earlier concerns for publication. I do want to restate, however, the point that I am NOT a fan of the term 'health controls' as this is presumptive and people in the 'matched non-pain

comparison group' were not comprehensively screened to exclude physical or psychological pathology. The sample seemingly had almost half above a 'healthy' BMI and perhaps one or two approached borderline 'unhealthy' alcohol intake.

Response: Thank you for pointing this out. The two groups are now named as 'no-pain' and 'chronic low back pain'.

Dear Dr Venezia,

Re: JP-RP-2024-286375R2 "Investigating the Effects of Artificial Baroreflex Stimulation on Pain Perception: A Comparative Study in No-pain and Chronic Low Back Pain Individuals" by Alessandra Venezia, Harriet Fawsitt-Jones, David Hohenschurz-Schmidt, Matteo Mancini, Matthew A Howard, and Elena Makovac

We are pleased to tell you that your paper has been accepted for publication in The Journal of Physiology.

IMPORTANT

We seem to be missing a legend for your abstract figure. Our typesetters will require this. Please can you email us the abstract figure legend as soon as possible, please? You can send it to: jp@physoc.org
Thank you!

Authors should note that it is too late at this point to offer corrections prior to proofing. Major corrections at proof stage, such as changes to figures, will be referred to the Editors for approval before they can be incorporated. Only minor changes, such as to style and consistency, should be made at proof stage. Changes that need to be made after proof stage will usually require a formal correction notice.

If you would like to receive our 'Research Roundup', a monthly newsletter highlighting the cutting-edge research published in The Physiological Society's family of journals (The Journal of Physiology, Experimental Physiology and Physiological Reports), please click this link, fill in your name and email address and select 'Research Roundup':
<https://www.physoc.org/journals-and-media/membernews/>.

Yours sincerely,

David Wyllie
Senior Editor
The Journal of Physiology

P.S. - You can help your research get the attention it deserves! Check out Wiley's free Promotion Guide for best-practice recommendations for promoting your work at www.wileyauthors.com/eoo/guide. You can learn more about Wiley Editing Services which offers professional video, design, and writing services to create shareable video abstracts, infographics, conference posters, lay summaries, and research news stories for your research at www.wileyauthors.com/eoo/promotion.

IMPORTANT NOTICE ABOUT OPEN ACCESS: To assist authors whose funding agencies mandate public access to published research findings sooner than 12 months after publication, The Journal of Physiology allows authors to pay an Open Access (OA) fee to have their papers made freely available immediately on publication.

You can check if your funder or institution has a Wiley Open Access Account here: <https://authorservices.wiley.com/author-resources/Journal-Authors/licensing-and-open-access/open-access/author-compliance-tool.html>.

EDITOR COMMENTS

Reviewing Editor:

Thank you for addressing these remaining comments. I am pleased to endorse acceptance of your paper.

Senior Editor:

Thank you for making these further revisions to your manuscript - I am happy to accept. Thank you for submitting this work to The Journal of Physiology.

2nd Confidential Review

30-Aug-2024